# BLISS: A Lightweight Bilevel Influence Scoring Method for Data Selection in Language Model Pretraining

Jie Hao [1]  Rui Yu [1]  Wei Zhang [2]  Huixia Judy Wang [3]  Jie Xu [4]  Mingrui Liu [1]

## Abstract

Effective data selection is essential for pretraining large language models (LLMs), improving efficiency and generalization to downstream tasks. However, existing approaches often rely on external pretrained models, making it difficult to separate the benefits of data selection from those introduced by external models. In addition, many methods estimate data importance from a fixed model state or short-horizon update, making it hard to capture how data preference changes as the model evolves during pretraining. In this paper, we introduce BLISS (**B**ileve**L** **I**nfluence **S**coring method for data **S**election), a lightweight data selection method that operates entirely *from scratch*, without external pretrained oracle models, while modeling dynamic data preference. BLISS uses a small proxy model as a surrogate for the LLM and trains a score model to estimate sample importance through multi-step proxy updates induced by score-weighted training data. We formulate data selection as a bilevel optimization problem: the upper-level objective optimizes the score model to assign sample weights, so minimizing the lower-level weighted training loss improves validation performance. Once optimized, the score model predicts influence scores, enabling efficient selection of high-quality samples for LLM pretraining. We validate BLISS by pretraining 410M/1B/2.8B Pythia and LLaMA-0.5B models on selected C4 subsets. Under the 1B setting, BLISS achieves a $1.7\times$ speedup in reaching the same performance as the state-of-the-art method, while delivering superior performance across multiple downstream tasks.

[1]Department of Computer Science, George Mason University, USA, [2]IBM T.J. Watson Research Center, USA, [3]Department of Statistics, Rice University, [4]Department of System Engineering & Operations Research, George Mason University, USA. Correspondence to: Mingrui Liu <mingruil@gmu.edu>.

*Proceedings of the 43$^{rd}$ International Conference on Machine Learning*, Seoul, South Korea. PMLR 306, 2026. Copyright 2026 by the author(s).

## 1. Introduction

The successful large-scale language model pretraining crucially relies on the careful choice of pretraining data (Brown et al., 2020; Raffel et al., 2020; Du et al., 2022; Elazar et al., 2023). Recent studies have shown that effective data selection (a.k.a., data curation) methods can enhance pretraining efficiency (Xie et al., 2023a) and improve generalization (Engstrom et al., 2024; Wettig et al., 2024). There are various types of data selection approaches for language model pretraining, including language filtering (Laurençon et al., 2022; Wenzek et al., 2019), data deduplication (Lee et al., 2021; Abbas et al., 2023), heuristic approaches (Rae et al., 2021; Penedo et al., 2023), data-quality filtering (Brown et al., 2020; Gao et al., 2020; Chowdhery et al., 2023; Xie et al., 2023b; Wettig et al., 2024), data mixing (Xie et al., 2023a; Albalak et al., 2023; Xia et al., 2023), and data influence function based methods (Park et al., 2023; Engstrom et al., 2024; Yu et al., 2024). Despite the rich literature of data selection methods in large language model (LLM) pretraining (e.g., a survey paper in Albalak et al. (2024)), it is still unclear what properties are needed for the training data curation to guarantee good performance: it remains an important real-world challenge (Li et al., 2024).

Existing approaches of data selection methods suffer from two major limitations. First, they often require leveraging pretrained models (Brown et al., 2020; Xie et al., 2023b; Wettig et al., 2024) for data-quality filtering, making it difficult to separate the effects of data selection from those of the external pretrained models. For example, the QuRating method (Wettig et al., 2024) assigns quality ratings to training samples based on responses from a pretrained LLM (e.g., GPT-3.5) before training a QuRater model. This reliance raises uncertainty about the role of the external LLM in the training process and whether its feedback is truly optimal. Moreover, the cost of invoking these external pretrained models is prohibitively expensive during data selection process for large-scale pretraining. Second, many influence-based approaches estimate data utility from a fixed model state or a short-horizon update. For example, the data influence function based approach (Yu et al., 2024) evaluates the impact of individual training samples based on a single training step with the current model, which does

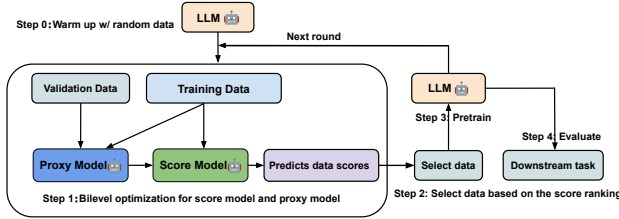

*Figure 1.* The pipeline of data selection and pretraining procedure. There are four main steps in one round training, 1) Warm up LLM using randomly selected training data (e.g. 10k step); 2) Bilevel optimization for score and proxy model, 3) Predict the data influence, and select Top-20% training data based on their score ranking; 4) Retrain the LLM using the selected data (e.g., 10k steps); 5) Evaluate on the downstream task. Repeating the above steps can achieve multiple-round training.

not capture the cumulative effects of data selection over the course of full model training.

In this paper, we introduce a new data selection method to address the two major limitations of existing approaches. Our method, namely BLISS (**B**ileve**L** **I**nfluence **S**coring method for data **S**election), is a lightweight data selection method that operates entirely *from scratch*, without relying on any external pretrained models, while explicitly modeling dynamic data preference under model updates. The core innovation of our approach lies in *the integration of two lightweight models within a novel bilevel optimization framework* for data selection. Our method bypasses traditional data-quality filtering and uses bilevel optimization to adapt sample weights according to the evolving proxy-model training dynamics. In particular, BLISS leverages a small proxy model as a surrogate for the LLM and employs a score model to estimate sample importance through score-induced proxy-model updates. Our bilevel optimization problem has upper-level and lower-level objectives: the upper-level objective optimizes the score model to assign importance weights to training samples, ensuring that minimizing the lower-level objective (i.e., training the proxy model over the weighted training loss until convergence) leads to best validation performance. Once the bilevel optimization is solved, the trained score model predicts influence scores for the entire dataset, enabling the selection of high-score samples for LLM pretraining. The pipeline of our proposed procedure is illustrated in Figure 1. The main contributions of our paper are summarized as the following:

- We propose a principled approach to data selection for language model pretraining. Our method, BLISS, leverages a novel bilevel optimization framework that employs a proxy model and a score model to model dynamic data preference under model updates. Unlike existing methods, BLISS operates from scratch without relying on any pretrained oracle models for data-quality filtering, obviating any biases or risks that may arise

from such dependence[1].

- We validate our method by pretraining 410M/1B Pythia and LLaMA-0.5B models on selected subsets of C4 dataset. Experimental results on 1B setting demonstrate a $1.7\times$ speedup in reaching the same performance as the state-of-the-art method such as MATES (Yu et al., 2024). Furthermore, we scale up our experiments to pretraining a 2.8B model using the data selected in the 1B experiment, and demonstrate that our method consistently outperforms MATES at every round of data selection, achieving $1.4\%$ performance improvement over MATES (Yu et al., 2024).

- Through extensive ablation studies, we demonstrate the effectiveness of each component in our bilevel optimization framework, further substantiating the robustness and efficiency of our approach.

## 2. Related Work

**Data Selection for Language Model Training.** Early approaches to data selection primarily relied on rule-based methods as language filters for training data, employing utility functions tailored to specific datasets (Conneau & Lample, 2019; Raffel et al., 2020; Rae et al., 2021; Penedo et al., 2023). Another key category is data deduplication (Lee et al., 2021; Sorscher et al., 2022; Penedo et al., 2023; Abbas et al., 2023; Tirumala et al., 2023), which eliminates redundant samples to optimize training efficiency and enhance performance on downstream tasks. A class of methods exist for performing data-quality filtering, which can select data similar to high-quality corpus of data points (Brown et al., 2020; Du et al., 2022; Gao et al., 2020; Xie et al., 2023b; Li et al., 2024), with small perplexity (Chowdhery et al., 2023; Wenzek et al., 2019). More recent methods leverage external pretrained LLMs to evaluate the pretraining data quality (Wettig et al., 2024; Maini et al., 2024; Zhuang et al., 2025). In addition, a similar variant of data selection is domain reweighting for data mixtures (Oren et al., 2019; Sagawa et al., 2019; Xie et al., 2023a; Fan et al., 2023; Albalak et al., 2023; Chen et al., 2023), which re-scale the contribution of each domain to enhance generalization. Another recently emerged line of research leverages the tool of influence functions (Hampel, 1974; Cook, 1977; Cook & Weisberg, 1982; Koh & Liang, 2017) to evaluate the impact of individual training samples on a fixed LLM (Park et al., 2023; Engstrom et al., 2024; Yu et al., 2024; Pan et al., 2025;

---

[1]Many commercial large-scale pretrained models strictly prohibit users from generating data or using them to facilitate the training of other models, as doing so may result in severe legal consequences (OpenAI, 2024; Google, 2024). Our approach is entirely free from such legal concerns. We rely solely on algorithmic advancements applied to a model trained from scratch, without any dependence on third-party pretrained large-scale models.

Lin et al., 2024). QUAD (Zhang et al., 2024) proposes an efficient framework incorporating the attention layers to estimate the influence scores. Different from token-level attribution methods such as (Lin et al., 2024), which retrieve influential training data for a given token prediction of an already trained or fixed LLM, BLISS targets sample-level data selection for pretraining from scratch. BLISS also differs from classical influence-estimation approaches such as DataInf (Kwon et al., 2024), which approximate the effect of upweighting or removing individual examples around a fixed fine-tuning state. Moreover, methods such as ALinFiK (Pan et al., 2025) estimate per-sample influence through training/test gradient interactions on the target LLM. In contrast, BLISS does not compute influence scores directly on the target LLM; instead, it learns a scoring model through lightweight proxy-based bilevel optimization, with a KL alignment term encouraging the proxy model to reflect the target model's data preference.

In contrast to these works, our method models dynamic data preference induced by multi-step proxy-model updates, rather than estimating influence only around a fixed model state. In addition, our method can train the model from scratch and does not need any extra information from any external pretrained models, making it a scalable and effective solution.

**Bilevel Optimization and Data Selection.** Bilevel optimization provides a powerful framework for modeling optimization problems with a nested structure (Bracken & McGill, 1973; Dempe, 2002). Recent research has focused on developing efficient bilevel optimization algorithms with strong theoretical guarantees (Ghadimi & Wang, 2018; Hong et al., 2023; Ji et al., 2021; Kwon et al., 2023; Dagréou et al., 2022; Chen et al., 2024; Grazzi et al., 2022; Hao et al., 2024; Gong et al., 2024a;b; 2025; Wu et al., 2026). This approach has been widely applied in various machine learning tasks, including meta-learning (Finn et al., 2017), hyperparameter optimization (Franceschi et al., 2018), and natural language processing (Somayajula et al., 2023; Grangier et al., 2023). For the application of data selection, bilevel optimization has been utilized for continual learning (Borsos et al., 2020; Zhou et al., 2022; Hao et al., 2023) and data reweighting in LLM fine-tuning (Pan et al., 2024; Shen et al., 2024). Our work is most closely related to SEAL (Shen et al., 2024), which focuses on selecting high-quality and safe data to fine-tune a pretrained LLM, with the goal of aligning the model with safety and ethical guidelines. However, our approach differs from SEAL in two key aspects: (1) Problem setting. While SEAL operates in a fine-tuning context, our objective is to select data for **pretraining** an LLM **from scratch**, aiming to improve downstream performance **without relying on any external pretrained models**. (2) Model update mechanism. SEAL utilizes the LoRA technique (Hu et al., 2022) to update both

the data selector and the LLM during fine-tuning. However, this approach is not directly applicable to our setting due to the following reasons. First, LoRA is only suitable for fine-tuning tasks but insufficient for full model pretraining. Second, their algorithm always updates the original large models directly, which is computationally expensive if all parameters are updated. In contrast, we propose a more efficient framework that introduces lightweight models (a score model and a proxy model) to guide data selection, while allowing full parameter updates within these smaller networks. To the best of our knowledge, our proposed bilevel influence scoring method is the first to leverage bilevel optimization techniques for data selection in LLM pretraining.

## 3. Preliminaries and Notations

Suppose that we have a large-scale training dataset $\mathcal{D}_{tr} = \{\xi_i \mid 0 \leq i \leq N-1\}$ and a downstream task $\mathcal{D}_{ds}$. The goal is to select a subset of training set, namely $\mathcal{D}_s \subset \mathcal{D}_{tr}$ and $|\mathcal{D}_s| = Q \leq N$, to pretrain a large language model with a specific training budget (e.g., limited FLOPs), such that the model can achieve high performance on the downstream task $\mathcal{D}_{ds}$. Generally, the downstream data is inaccessible during pretraining. Instead, we can use a validation data $\mathcal{D}_{val} = \{\zeta_i \mid 0 \leq i \leq M-1\}$ to estimate the model's performance on $\mathcal{D}_{ds}$, because these two datasets often have similar data distributions or share common domain knowledge. A small subset of training data $\tilde{\mathcal{D}}_{tr} \subset \mathcal{D}_{tr}$ is uniformly sampled from $\mathcal{D}_{tr}$.

In bilevel optimization, $f(\cdot)$ and $g(\cdot)$ denote the upper-level (UL) and lower-level (LL) functions, respectively. Machine learning often requires solving stochastic optimization problems $f(\cdot) = \mathbb{E}_{\xi \sim \mathcal{D}_f}[F(\cdot; \xi)]$ and $g(\cdot) = \mathbb{E}_{\zeta \sim \mathcal{D}_g}[G(\cdot; \zeta)]$, where $\mathcal{D}_f$ and $\mathcal{D}_g$ are the underlying unknown data distribution for $f$ and $g$, respectively. $F(\cdot; \xi)$ denotes the upper-level stochastic objective function and $G(\cdot; \zeta)$ is the lower-level stochastic objective function. Noisy observations of $f$ and $g$ can be collected by sampling from $\mathcal{D}_f$ and $\mathcal{D}_g$.

## 4. Methods

### 4.1. Bilevel Influence Scoring Framework

The goal of data selection is to optimize the performance of the LLM on downstream tasks by training it using an optimal subset of training data. However, directly searching for the optimal subset of training samples faces prohibitive costs due to the combinatorial nature of the problem and the high computational cost of estimating the performance of the LLM for every potential subset being evaluated.

To address the aforementioned computational challenge, our bilevel influence scoring framework uses a lightweight score model $\theta_s$ to predict the influence of every sample on the

model's performance for the downstream task. The optimized score model is then used to infer the influence score of training samples, enabling the selection of the subset with the highest influence, thus streamlining the process to search for the optimal training data. Instead of directly estimating the performance of LLM (parameterized by $\theta_{tr}$) which is computationally expensive, our framework introduces a lightweight proxy model $\theta_p$ to approximate the behavior of the LLM. Note that the score model and the proxy model are both small models: they share a similar architecture and number of parameters. To ensure the data preferences of the proxy model align with those of the LLM, we apply knowledge distillation by minimizing the Kullback-Leibler (KL) divergence between the output logits of the proxy model and the LLM. We formulate the bilevel optimization for data selection as follows:

$$\min_{\theta_s} \Phi(\theta_s) := f(\theta_p^*(\theta_s)) := \mathbb{E}_{\zeta \sim \mathcal{D}_{\text{val}}} F(\theta_p^*(\theta_s); \zeta) \quad \text{(UL)},$$

$$\text{s.t. } \theta_p^*(\theta_s) = \arg\min_{\theta_p} g(\theta_p, \theta_s) := \mathbb{E}_{\xi \sim \mathcal{D}_{tr}} G(\theta_p, \theta_s; \xi)\text{(LL)},$$

$$(1)$$

where $\mathbb{E}_{\xi \sim \mathcal{D}_{tr}} G(\theta_p, \theta_s; \xi) = \sum_{i=0}^{N-1} P_i \mathcal{L}(\theta_p; \xi_i) + \gamma D_{KL}\left(\ell(\theta_p; \xi_i) \| \ell(\theta_{tr}; \xi_i)\right) + \lambda \|\theta_p\|^2$ and $P_i = \frac{e^{h(\theta_s; \xi_i)}}{\sum_{j=1}^{N} e^{h(\theta_s; \xi_j)}}$ represents the importance weight of sample $i$, and $h(\cdot) : \mathbb{R}^{d_x} \to (0, 1)$ is a function that maps a sample from $\mathbb{R}^{d_x}$ to an influence score in the range $(0, 1)$. $\mathcal{L}(\cdot)$ and $F(\cdot)$ denote the loss functions for next token prediction, with a common choice being cross-entropy. The normalized model's output logits are represented by $\ell(\cdot)$. The KL divergence is defined as $D_{KL}(X \| Y) = \sum_i X_i \log(\frac{X_i}{Y_i})$. $\gamma$ and $\lambda$ are the regularization coefficients for the KL divergence and the weight decay terms, respectively.

**Intuition of Algorithm Design.** The lower level asks: *given a candidate weighting over training samples, what proxy model will be obtained after training under these weights?* It updates the proxy model on the score-weighted training loss, together with a KL alignment term that encourages the proxy to reflect the data preference of the target LLM. The upper level then asks: *do these weights lead to better validation performance?* It updates the score model so that the resulting proxy model $\theta_p^*(\theta_s)$ achieves lower validation loss. In this way, the score model proposes sample weights, the proxy model reflects their effect through multi-step training dynamics, and the validation loss provides feedback to improve the score model.

This differs from one-step influence methods such as MATES (Yu et al., 2024), which estimate sample utility from a local update at the current model state. BLISS instead learns sample weights through a bilevel interaction between the score model and a dynamically updated proxy model, allowing the selected data to better reflect the evolv-

ing model state. Moreover, the framework in equation 1 does not rely on external pretrained oracle models, making BLISS a self-contained data selection approach for pretraining from scratch.

## 4.2. Algorithm for Updating the Proxy Model and Score Model

Now we design efficient algorithms for solving the bilevel problem (1). The lower-level problem aims to optimize the proxy model $\theta_p$ on the weighted training samples with the influence predicted by the score model. Note that we freeze the LLM ($\theta_{tr}$) through the process of solving the bilevel optimization problem, as the LLM is used to infer the output logits. Therefore, we perform the following update for the lower-level objective on a mini-batch of size $\mathcal{B}$ (full batch is infeasible in practice):

$$\theta_p^{t+1} = \theta_p^t - \eta_1 \nabla_{\theta_p} \sum_{i=1}^{\mathcal{B}} G(\theta_p^t, \theta_s^t; \xi_i)$$

$$= \theta_p^t - \eta_1 \sum_{i=1}^{\mathcal{B}} \left( P_i \nabla_{\theta_p} \mathcal{L}(\theta_p^t; \xi_i) \right. \quad (2)$$

$$\left. + \gamma \sum_j \nabla_{\theta_p} \ell_j(\theta_p^t; \xi_i) \log \frac{\ell_j(\theta_p^t; \xi_i)}{\ell_j(\theta_{tr}; \xi_i)} \right) - 2\eta_1 \lambda \theta_p^t,$$

where $\ell_j(\cdot)$ denotes the $j$-th normalized logit of the output. Note that the exact computation of $P_i$ depends on all $N$ samples, which is computationally infeasible. Therefore, we approximate $P_i$ by replacing the full summation in the denominator with a partial summation over a smaller subset. This approximation is implemented in a distributed manner, significantly reducing the computational overhead. More details can be found in Appendix H. For the upper-level update, we take the derivative of $\Phi(\theta_s)$ with respect to $\theta_s$ by chain rule, which is known as the hypergradient:

$$\nabla_{\theta_s} \Phi(\theta_s)$$
$$= -\nabla^2_{\theta_s \theta_p} g(\theta_p^*(\theta_s), \theta_s) \underbrace{[\nabla^2_{\theta_p} g(\theta_p^*(\theta_s), \theta_s)]^{-1} \nabla_{\theta_p} f(\theta_p^*(\theta_s))}_{z},$$
$$(3)$$

where $z$ is the solution of the quadratic function $\min_z \frac{1}{2} z^T \nabla^2_{\theta_p} g(\theta_p^*(\theta_s), \theta_s) z - z^T \nabla_{\theta_p} f(\theta_p^*(\theta_s))$. It can be solved by running a few steps of gradient descent in practice:

$$z_{k+1}^t = z_k^t - \eta_2 \left( \nabla^2_{\theta_p} g(\theta_p^t, \theta_s^t) z_k^t - \nabla_{\theta_p} f(\theta_p^t) \right), \quad (4)$$

where $k$ is the number of gradient updates for updating $z$ at a fixed iteration $t$ of updating $\theta_s$. We run 3 steps of gradient descent to solve $z$ in our experiments. Note that Equation (4) computes the Hessian-Vector-Product (HVP) term $\nabla^2_{\theta_p} g(\theta_p^t, \theta_s^t) z_k^t$ and thus avoids the computationally

prohibitive operation of taking the inverse of the Hessian. The dimension of $z$ is the same as that of the parameters of the lightweight proxy model. Therefore, the computation of HVP within the PyTorch framework is quite similar to that of gradient. In our implementation, we use the stochastic variants of Equation (3) and Equation (4) for updating the score model. In particular, the approximation of hypergradient at iteration $t$ on the mini-batch $\mathcal{B}$ is

$$\nabla_{\theta_s} \widehat{\Phi}(\theta_s^t) = -\sum_{i=1}^{\mathcal{B}} P_i \nabla_{\theta_s} h(\theta_s^t; \xi_i) \nabla_{\theta_p} \mathcal{L}(\theta_p^t; \xi_i)^T z^t$$
$$+ \sum_{i=1}^{\mathcal{B}} P_i \sum_{j=1}^{\mathcal{B}} P_j \nabla_{\theta_s} h(\theta_s^t; \xi_j) \nabla_{\theta_p} \mathcal{L}(\theta_p^t; \xi_i)^T z^t. \quad (5)$$

Then the update for the upper-level variable ($\theta_s$) is $\theta_s^{t+1} = \theta_s^t - \alpha \nabla_{\theta_s} \widehat{\Phi}(\theta_s^t)$. When the score model converges over $T$ steps, reaching $\theta_s^T$, it is then used to estimate the influence scores of the entire training dataset in the current round by: $S_i = h(\theta_s^T, \xi_i), \forall \xi_i \in \mathcal{D}_{tr}$. Then the influence scores are collected: $\{S_i \mid 0 \leq i \leq |D_{tr}|\}$, and the top-ranked samples with the highest influence scores are selected to construct $\mathcal{D}_s$, which is used to pretraining the LLM ($\theta_{tr}$).

The detailed implementation of the algorithm is presented in Algorithm 1. In practice, we call SGD optimizer SGD(variable, gradient, lr, steps) to update the lower-level variable for $N$ steps with learning rate lr. We use Adam optimizer (Adam(variable, gradient, lr)) (Kingma & Ba, 2015) to update the upper-level variables. The pretraining process is conducted over $R$ rounds. In each round, the algorithm performs data selection followed by LLM retraining. The training dataset is partitioned into $R$ shards. The data selection in round $r$ is conducted on $\mathcal{D}_{tr}^r$. The LLM resumes training from the previous round's checkpoint and updates to $\theta_{tr}^r$ at the end of the $r$-th round. Similarly, the score model also continues learning throughout the process, reaching $\theta_s^r$ at the $r$-th round.

It is worth noting that the proxy model $\theta_p$ is reinitialized with the warm-up proxy at the beginning of each round, while the target LLM continues training from the previous round. The data distribution changes substantially from one round to the next (i.e., $\xi$ comes from a different data distribution when the round changes), which leads to a completely different lower-level problem. As a result, simply warm-starting as (Ji et al., 2021) from the proxy of the previous round may introduce a strong bias toward the previous selected distribution. In our setting, this can reduce the proxy's ability to adapt to the current round, i.e., its *plasticity*. This intuition is consistent with prior work (Shin et al., 2024) on loss of plasticity, which shows that under non-stationary training distributions, a network trained for a long time can become harder to adapt than a freshly initialized counterpart.

---

**Algorithm 1** BLISS

1: **Input:** $\eta_1, \eta_2, \eta_3, \alpha, R, T, K, Q, N, \mathcal{D}_{tr}, \tilde{\mathcal{D}}_{tr}, \mathcal{D}_{val}$
2: **Initialize:** Warm up $\theta_p^{0,0}, \theta_s^{0,0}, \theta_{tr}^{0,0}$ using randomly selected training data.
3: **for** $r = 0, \dots, R-1$ **do**
4:   # *reset proxy/score parameters for a new round*
5:   $\theta_p^{0,r} = \theta_p^{0,0}$
6:   $\theta_s^{0,r} = \theta_s^{T,r-1}$ if $r \geq 1$ else $\theta_s^{0,0}$
7:   $\theta_{tr}^{0,r} = \theta_{tr}^{Q,r-1}$ if $r \geq 1$ else $\theta_{tr}^{0,0}$
8:   **for** $t = 0, \dots, T-1$ **do**
9:     Sample $\xi_t^r, \tilde{\xi}_t^r, \pi_t^r \leftarrow \tilde{\mathcal{D}}_{tr}^r$, and sample $\zeta_t \leftarrow \mathcal{D}_{val}$
10:     # *LL: update the proxy model for N steps*
11:     $\theta_p^{t+1,r} = \text{SGD}(\theta_p^{t,r}, \nabla_{\theta_p} G(\theta_p^{t,r}, \theta_s^{t,r}; \xi_t^r), \eta_1, N)$
12:     # *solve the linear system*
13:     $z^{t+1,r} = \text{GDLS}(\eta_2, K, \nabla_{\theta_p} G(\theta_p^{t,r}, \theta_s^{t,r}; \tilde{\xi}_t^r),$
                 $\nabla_{\theta_p} F(\theta_p^{t,r}, \theta_s^{t,r}; \zeta_t))$
14:     # *UL: update the score model*
15:     $\theta_s^{t+1,r} = \text{Adam}(\theta_s^{t,r}, -\nabla_{\theta_s \theta_p}^2 G(\theta_p^{t+1,r}, \theta_s^{t,r};$
                 $\pi_t^r) z^{t+1,r}, \alpha)$
16:   **end for**
17:   Infer the influence score $\{S_i^r \mid 0 \leq i \leq |\mathcal{D}_{tr}^r| - 1\}$ on $\mathcal{D}_{tr}^r$ using $\theta_s^{T,r}$
18:   Sort $\{S_i^r\}$ in descending order and select the 20% data with the highest influence scores from $\mathcal{D}_{tr}^r$ to form the selected data $\mathcal{D}_s$
19:   **for** $\tau = 0, \dots, Q-1$ **do**
20:     Sample $\xi_\tau$ from $\mathcal{D}_s$.
21:     # *pretrain the LLM*
22:     $\theta_{tr}^{\tau+1,r} = \theta_{tr}^{\tau,r} - \eta_3 \nabla_{\theta_{tr}} \mathcal{L}(\theta_{tr}^{\tau,r}; \xi_\tau)$
23:   **end for**
24: **end for**

---

**Algorithm 2** GDLS: Gradient Descent for the Linear System Solution

1: **Input:** $\eta, K, \nabla_{\theta_p} g(\theta_p), a$
2: **Initialize:** $z_0$
3: **for** $k = 0, \dots, K-1$ **do**
4:   $z_{k+1} = z_k - \eta(\nabla_{\theta_p}^2 g(\theta_p) z_k - a)$
5: **end for**
6: Return $z_K$

---

Therefore, resetting the proxy $\theta_p$ with random initialization at each round is intended to let it re-fit the newly data distribution with high plasticity and keep alignment with the target LLM, rather than inheriting optimization bias from earlier rounds.

### 4.3. Warm Up Models

The key distinction between our algorithm and other data selection methods (Brown et al., 2020; Xie et al., 2023b; Wettig et al., 2024) is that it operates independently of exter-

nal pretrained models, avoiding biases from data selection influenced by such models. However, without leveraging pretrained knowledge, the proxy model, score model, and LLM tend to perform poorly in the initial phase due to random parameter initialization. To mitigate this issue, we incorporate a model warm-up step before data selection, similar to other data selection approaches (Yu et al., 2024; Xia et al., 2024), using randomly selected samples. The lightweight proxy and score models share token embedding layers and transformer blocks but differ in their final layers: the proxy model handles token generation, while the score model outputs influence scores for individual samples. Consequently, only the proxy model and the LLM require warm-up, while the score model can be initialized with the weights from proxy model directly.

We provide an additional convergence analysis of our algorithm in Appendix A, which clarifies the role of the proxy update steps, the selection ratio, and the round-wise proxy reset.

# 5. Experiments

In this section, we validate the proposed bilevel influence scoring framework for pretraining data selection. We apply the bilevel optimization algorithm to train a lightweight proxy model ($\theta_p$) and a score model ($\theta_s$)) for data selection. We then pretrain a target LLM ($\theta_{tr}$), specifically Pythia-410M/1B, from scratch on a selected subset of the large-scale C4 dataset (Raffel et al., 2020), which is designed for LLM pretraining. we then evaluate the pretrained LLM on multiple downstream tasks and compare its performance against several baseline methods, including Random selection, DSIR (Xie et al., 2023b), SemDeDup (Abbas et al., 2023), DsDm (Engstrom et al., 2024), LESS (Xia et al., 2024), QuRating (Wettig et al., 2024), and MATES (Yu et al., 2024). We furthermore scale up our experiment to 2.8B model pretraining and achieve $1.4\%$ performance improvement over the state-of-the-art method. We also verify the the fidelity of proxy models to full-scale LLMs by domain reweighting experiment on SlimPajama-6B, deferred to Appendix K. The code is available at `https://github.com/MingruiLiu-ML-Lab/BLISS-Bilevel-Data-Selection`.

## 5.1. Dataset Settings

Following the approach of DsDm (Engstrom et al., 2024), we perform data selection and pretraining using tokenized data. The procedure of BLISS is implemented for 5 rounds (i.e., $R = 5$),with the C4 dataset partitioned into five equal shards, denoted as $\{\mathcal{D}_{tr}^r \mid 0 \leq r \leq 4\}$. Each training round operates on a distinct data shard without replacement. In every round, we first uniformly sample a small proportion ($0.1\%$) from $\mathcal{D}_{tr}^r$ as the bilevel training set $\tilde{\mathcal{D}}_{tr}^r$ for updat-

ing the proxy model. We choose LAMBADA (Paperno et al., 2016) as validation data for updating the score model as the prior work (Engstrom et al., 2024; Yu et al., 2024). LAMBADA requires broad discourse context for word prediction and therefore provides a semantically demanding upper-level signal beyond local next-token prediction. More details about the choice of validation set can be found in Appendix D.4. Other datasets, including ARC-E (Clark et al., 2018), SQUAD (Rajpurkar et al., 2016), and PIQA (Bisk et al., 2020), are evaluated in the ablation study (Appendix D.4).

To evaluate the performance of data selection algorithms, we run the pretraining model across 9 downstream tasks, including SciQ (Welbl et al., 2017), ARC-E (Clark et al., 2018), ARC-C (Clark et al., 2018), LogiaQA (Liu et al., 2020), OBQA (Mihaylov et al., 2018), BoolQ (Clark et al., 2019), HellaSwag (Zellers et al., 2019), PIQA (Bisk et al., 2020), and WinoGrande (Sakaguchi et al., 2021). These tasks cover a diverse range of reasoning and comprehension challenges, including question answering, logical inference, commonsense reasoning, and coreference resolution. Thus it requires models to demonstrate various capabilities, such as retrieving and applying scientific knowledge, understanding causal relationships, resolving ambiguities in natural language, and making informed choices among distractors. A good data selection algorithm is expected to select the "important" data that boost model performance across these downstream tasks.

## 5.2. Model Settings

The target pretraining model, Pythia-410M/1B/2.8B, consists of 410 million, 1 billion or 2.5 billion trainable parameters. Both the proxy model and score model are based on Pythia-31M (for Pythia-410M) or Pythia-160M (for Pythia-1B), but they serve different purposes: the proxy model acts as a surrogate for the LLM and is trained for next-token prediction, while the score model functions as a regression model that maps individual samples to corresponding influence scores. Details of model settings are deferred to Appendix B. Notably, all models are trained from scratch using Gaussian initialization for model parameters. Additional experimental details, including hyperparameter choices, learning rate schedules, and distributed training strategies, are provided in Appendix F.

## 5.3. Bilevel Optimization for Proxy Model and Score Model

In the Pythia-410M setting, the proxy model $\theta_p$ is updated with a "single-step" ($N = 1$) optimization per iteration (line 9 in Algorithm 1). However, when scaling up to larger models like Pythia-1B, we adopt a "multi-steps" ($N = 5$) update strategy for the proxy model to achieve a better lower-

level solution. To demonstrate the effectiveness of bilevel optimization in training the proxy model and score model, we visulize the evolution of the training loss at during round 2 (Figure 6a in Appendix G) and round 5 (Figure 6b in Appendix G). Since the first round uses randomly selected data to warm up the LLM, our data selection algorithm is employed from the second round onward.

Within each round, both losses exhibit a two-phase trend: they initially decrease rapidly before experiencing a slight increase. This behavior arises due to the composition of the lower-level objective function, which includes three terms: the weighted cross-entropy loss, the KL divergence loss, and a regularization term. In the first phase, the weighted cross-entropy loss dominates, decreasing as the proxy model is optimized. In the second phase, the KL divergence term becomes more influential. Since the LLM has not yet been trained on the current dataset $\mathcal{D}_{tr}^r$ (it only performs inference in bilevel training), its predictions may be suboptimal. The KL divergence term encourages the proxy model to mimic the behavior of this "imperfect" LLM, leading to a slight performance degradation. However, this ensures that the proxy model's data preference aligns with that of the LLM, improving the relevance of the selected training data and ultimately boosting the LLM's downstream task performance. An ablation study on the effect of KL divergence loss is presented in Section 6.2.

From round 2 to round 5, the score model is continuously optimized, leading to more accurate sample weight assignments. This, in turn, enhances the proxy model's performance on the weighted training samples, further improving the quality of data selection.

### 5.4. Evaluation Results on the Downstream Tasks

The LLM is continuously trained for 10,000 steps on the selected data in each round. After completing five rounds of training, we evaluate the zero-shot performance of Pythia-410M/1B on various downstream tasks and report the average accuracy along with the standard error for each dataset(see Table 1. Our algorithm consistently outperforms MATES and random selection methods across multiple tasks. For example on 410M setting, BLISS, compared with MATES, improves $2.4\%$ on SciQ, $0.7\%$ on ARC-E, $0.8\%$ on LogiQA, $0.2\%$ on HellaSwag, $0.4\%$ on PIQA, $0.2\%$ on WinoGrande, and $0.2\%$ on average accuracy (see Table 8). Additionally, Figure 2 presents the evaluation results in relation to pretraining FLOPs and training steps. BLISS consistently outperforms other baseline methods throughout the entire five-round pretraining process (with 10k steps per round). In particular, our method on 1B setting achieves a $1.7\times$ speedup in reaching the same performance as MATES, further validating the effectiveness of our data selection approach.

**Scaling Up to 2.8B Model Pretraining using the Data Selected by 160M/1B Experiment.** To further validate the selected data is of good quality regardless of model size, we pretrain a larger model of 2.8B parameters with data selected from the 1B model experiment with 160M proxy and score models. We run MATES and BLISS for 3 rounds (15B tokens). As shown in Table 2, BLISS consistently outperforms MATES across all data selection rounds, achieving $1.4\%$ accuracy improvement over MATES in round 3.

**Generalize model architecture to LLaMA family.** We also explore LLaMA architecture models to validate the generalization of our method. Specifically, we use LLaMA-0.5B as the target pretraining model, and LLaMA-134M as the proxy model and score model. In each round, we first minimize the difference between the proxy model and the target model by training the proxy model toward a lower KL divergence. Then we periodically reset the proxy model to the initial state, in addition to resetting it at the beginning of each round. Table 3 presents the evaluation results compared with MATES, where BLISS exhibits strong data selection performance. At the round 3, our algorithm improves over MATES by $0.6\%$. More details are presented in Appendix C.

### 5.5. Computational Cost

We follows the FLOPs estimation method in Li et al. (2024) and report the total GPU FLOPs, including the pretraining, model warm-up, and data selection. Our main observation is: without relying on any external pretrained models as required in MATES, BLISS achieves higher average downstream performance while consuming fewer FLOPs. A detailed comparison of total FLOPs consumption is provided in Table 4, and running time/memory comparison is presented in Appendix I.

With the same pretraining budget for LLM and an equivalent number of training tokens, BLISS is more efficient in data selection than MATES. The higher computational cost of MATES is due to its reliance on oracle data influence estimation, which involves computing the loss change after performing a one-step gradient descent update on an individual training sample. This process is highly time-consuming, because it requires per-sample gradient and cannot increase the batch size per GPU. In contrast, BLISS formulates data selection as a bilevel optimization problem, enabling the lightweight score model and proxy model to be trained to convergence within relatively few steps, i.e., 3,000 per round (ablation study for bilevel steps is presented in Appendix J). While BLISS introduces additional training steps for warming up the proxy and score models from scratch, this cost is negligible compared to the overall pretraining FLOPs. One limitation of BLISS is that the HVP computation in bilevel optimization incurs higher peak memory usage than

*Table 1.* Comparison of methods on zero-shot evaluation over multiple downstream datasets (410M/1B model, 25B tokens data). Best results are marked bold. The accuracy with standard error is reported based on the lm-evaluation-harness (Gao et al., 2021) implementation.

| Methods (#FLOPs $\times 10^{19}$) | SciQ | ARC-E | ARC-C | LogiQA | OBQA | BoolQ | HellaSwag | PIQA | WinoGrande | Average |
|---|---|---|---|---|---|---|---|---|---|---|
| **410M Setting:** 410M model, 25B tokens | | | | | | | | | | |
| Random (6.35) | 64.1 (1.5) | 40.2 (1.0) | **25.6** (1.3) | 24.7 (1.7) | 29.4 (2.0) | 58.9 (0.9) | 39.7 (0.5) | 67.1 (1.1) | 50.6 (1.4) | 44.5 (1.3) |
| MATES (8.11) | 65.7 (1.5) | 41.5 (1.0) | 25.0 (1.3) | 26.1 (1.7) | **30.8** (2.1) | **60.6** (0.9) | 41.0 (0.5) | 67.8 (1.1) | 51.8 (1.4) | 45.7 (1.4) |
| BLISS (8.08) | **68.1** (1.5) | **42.2** (1.0) | 25.1 (1.3) | **27.3** (1.7) | 29.6 (2.0) | 59.3 (0.9) | **41.2** (0.5) | **68.2** (1.1) | **52.0** (1.4) | **45.9** (1.4) |
| **1B Setting:** 1B model, 25B tokens | | | | | | | | | | |
| Random (17.67) | 65.8 (1.5) | 43.7 (1.0) | 25.6 (1.3) | 27.5 (1.8) | 31.8 (2.1) | 60.2 (0.9) | 43.8 (0.5) | 68.9 (1.1) | 50.7 (1.4) | 46.4 (1.4) |
| MATES (19.97) | 67.3 (1.5) | 44.9 (1.0) | **25.9** (1.3) | **28.7** (1.8) | 32.2 (2.1) | **60.9** (0.9) | 45.3 (0.5) | 69.5 (1.1) | 52.4 (1.4) | 47.5 (1.4) |
| BLISS (19.53) | **69.4** (1.5) | **45.7** (1.0) | 24.8 (1.3) | 25.8 (1.7) | **33.2** (2.1) | 59.8 (0.9) | **47.8** (0.5) | **71.6** (1.1) | **52.9** (1.4) | **47.9** (1.3) |

*Table 2.* Average evaluation accuracy (15B tokens) by pretraining the 2.8B model with data selected from the 1B-model experiment.

| Methods | Round 1 (Random) | Round 2 | Round 3 |
|---|---|---|---|
| MATES | 45.9 (1.3) | 47.4 (1.3) | 47.6 (1.3) |
| BLISS | 45.2 (1.3) | **47.6** (1.3) | **49.0** (1.3) |

*Table 3.* Average evaluation accuracy over three rounds by pretraining the LLaMA-0.5B model. LLaMA-134M is used as the proxy model in BLISS.

| Methods | Round 1 (Random) | Round 2 | Round 3 |
|---|---|---|---|
| MATES | 43.12 (1.27) | 44.53 (1.27) | 45.01 (1.27) |
| BLISS | 43.12 (1.27) | **44.57** (1.27) | **45.65** (1.27) |

MATES, as reported in Table 10. This represents a memory-time trade-off: BLISS requires more peak memory, but its data-selection stage is substantially faster because it avoids per-sample one-step influence estimation on the target LLM.

## 6. Ablation Studies

To inspect the effectiveness of key techniques used in our proposed algorithm, we conduct ablation studies on the effect of bilevel optimization (Section 6.1), KL divergence loss (Section 6.2), the impact of softmax reparameterization on the score model's outputs (Appendix D.1), the size of proxy model (Appendix D.2), the initialization for the score model (Appendix D.3), and the influence of different validation datasets ($\mathcal{D}_{val}$) on performance (Appendix D.4).

### 6.1. Single-level versus Bilevel Optimization

In bilevel algorithm, the hyper-gradient is essential for the update of upper level parameters. To verify the effectiveness of bilevel update for the upper-level parameters, we compare bilevel update with a single update, which update $\theta_s$ and $\theta_p$ together using both training and validation data for the lower-level objective. Specifically, the upper and lower levels are reduced to a single level problem: the upper-level

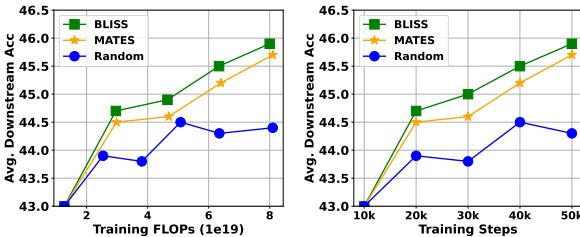

*(a)* Pretraining the 410M model.

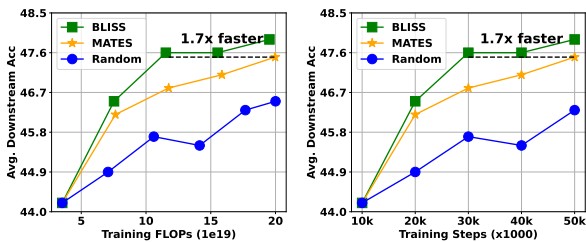

*(b)* Pretraining the 1B model.

*Figure 2.* Downstream performance of Pythia-410M and Pythia-1B with respect to pretraining FLOPs and steps. The first point corresponds to a warm-up model trained on random data.

and lower-level parameters are updated simultaneously on validation dataset and training dataset respectively. With the same number of training steps as bilevel training, the average accuracy of single level update degrades $0.5\%$ as shown in Table 5 in Appendix D.

### 6.2. Alignment via KL Divergence

Our objective is to select training data that maximizes the LLM's performance on downstream tasks. To achieve this, the proxy model must effectively represent the LLM, which we enforce by applying KL divergence loss to align their output logits. As shown in Figure 3 (Appendix D), incorporating KL divergence leads to improved performance across most downstream tasks, with a 9.3% accuracy boost on LogiQA and a 1.4% increase in average accuracy. Interestingly, while removing KL divergence results in a lower training loss (as seen in Figure 7 compared to Figure 6a in Appendix G), it does not translate to better downstream

*Table 4.* Total FLOPs for pretraining 410M/1B model with 25B tokens.

| Model | #FLOPs $\times 10^{19}$ | Ratio | Model | #FLOPs $\times 10^{19}$ | Ratio |
|---|---|---|---|---|---|
| **MATES:** 410M model, 25B tokens | | | **BLISS:** 410M model, 25B tokens | | |
| Model pretraining | 6.35 | 78.3% | Model pretraining | 6.35 | 78.59% |
| Oracle data influence collection | 0.29 | 3.58% | Warm up the proxy/score model | 0.07 | 0.87% |
| Data influence model training | 0.01 | 0.1% | Bilevel optimization | 0.13 | 1.62% |
| Data influence model inference | 1.46 | 18.0% | Data influence model inference | 1.53 | 18.94% |
| **Total** | 8.11 | 100.00% | **Total** | **8.08** | 100.00% |
| **MATES:** 1B model, 25B tokens | | | **BLISS:** 1B model, 25B tokens | | |
| Model pretraining | 17.67 | 88.5% | Model pretraining | 17.67 | 90.48% |
| Oracle data influence collection | 0.83 | 4.1% | Warm up the proxy/score model | 0.07 | 0.36% |
| Data influence model training | 0.01 | 0.1% | Bilevel optimization | 0.261 | 1.34% |
| Data influence model inference | 1.46 | 7.3% | Data influence model inference | 1.53 | 7.83% |
| **Total** | 19.97 | 100.00% | **Total** | **19.53** | 100.00% |

performance. These findings highlight the importance of bridging the gap between the proxy model and the LLM to ensure effective data selection, demonstrating that a closer alignment between the two models leads to better overall performance.

## 7. Conclusion

In this paper, we present BLISS, a lightweight bilevel influence scoring method for data selection in language model pretraining. BLISS utilizes a proxy model, a score model, and a novel bilevel optimization framework to model dynamic data preference under model updates without relying on external pretrained models. Experimental results demonstrate its effectiveness in selecting data for pretraining Pythia and LLaMA models. However, the bilevel optimization in our method may incur higher peak memory usage, which remains an important direction for future work.

## Acknowledgements

This work is supported by NSF award #2436217, #2425687, #2613303, #2228603, #2411248. We gratefully acknowledge the computational resources provided by the IBM T.J. Watson Research Center. We also acknowledge the computational support from the NSF NAIRR project under NAIRR award #250070.

## Impact Statement

This paper presents work whose goal is to advance the field of machine learning. There are many potential societal consequences of our work, none of which we feel must be specifically highlighted here.

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

# A. Convergence Analysis and Practical Design Choices

Classical bilevel convergence analyses typically rely on a strongly-convex lower-level problem, so that the lower-level solution mapping is well-defined and smooth. This assumption also underlies many recent single-loop bilevel methods that update the upper-level and lower-level variables with the same frequency (Dagréou et al., 2022). However, this formulation is not necessarily suitable for BLISS. In our setting, the lower-level corresponds to training a proxy language model under score-induced sample weights, which may not be strongly convex.

To characterize the update rule of BLISS, we adopt a finite-horizon view and define the lower level through $N$ steps of stochastic gradient descent (SGD) on the proxy model:

$$
\begin{aligned}
\min_{\theta_s} \quad & \Phi_N(\theta_s) := \mathbb{E}_{\zeta \sim \mathcal{D}_{\text{val}}} \big[ F(\theta_p^N(\theta_s); \zeta) \big], \qquad \text{(UL)} \\
\text{s.t.} \quad & \theta_p^N(\theta_s) = \texttt{SGD}(\theta_p, \nabla_{\theta_p} G(\theta_p, \theta_s; \xi), \eta_1, N) \qquad \xi \sim \mathcal{D}_{\text{tr}}. \qquad \text{(LL)}
\end{aligned}
\tag{6}
$$

Here, given $\theta_s$, the proxy model is trained for $N$ SGD steps with step size $\eta_1$, and the upper-level objective is evaluated on the resulting finite-horizon proxy model. This surrogate objective $\Phi_N$ is faithful to BLISS, since each upper-level update interacts with a proxy model obtained under a fixed computational budget, rather than with an idealized exact lower-level optimum.

We next introduce a finite-horizon approximation assumption, which plays the same role as the lower-level accuracy condition in standard inexact bilevel analysis. Let $\theta_p^*(\theta_s)$ denote the unique optimal proxy-training target induced by the score model, and define the $N$-step lower-level approximation error at outer iteration $t$ as

$$
\delta_t^2 := \sup_{\theta_s^t} \mathbb{E} \left[ \left\| \theta_p^N(\theta_s^t) - \theta_p^*(\theta_s^t) \right\|^2 \right].
$$

We follow the assumptions in (Salehi et al., 2025) to derive the convergence rate, including function smoothness and differentiable Assumption 3.1, and step size Assumption 3.3 in (Salehi et al., 2025). Let $\nabla \Phi_N(\theta_s)$ be the upper-level stochastic gradient used by BLISS and let $\Phi(\theta_s)$ denote the ideal target objective. As in (Salehi et al., 2025), we assume the upper-level stochastic gradient

$$
\nabla \Phi_N(\theta_s^t) = \nabla \hat{\Phi}(\theta_s^t) + e_t(\theta_s^t),
$$

where $\mathbb{E}[\nabla \hat{\Phi}(\theta_s^t)] = \nabla \Phi(\theta_s^t)$, and $e_t(\theta_s^t)$ is the finite-horizon gradient error satisfying

$$
\mathbb{E}\|e_t(\theta_s^t)\|^2 \leq \delta_t^2.
$$

Thus, $\delta_t$ measures the maximal finite-horizon lower-level error, while $e_t(\theta_s^t)$ denotes the corresponding inexact hypergradient error. In each iteration $t$, the upper level variable is updated by one-step inexact stochastic gradient descent over $\theta_s$ (i.e., $\theta_s^{t+1} = \theta_s^t - \alpha_t \nabla \Phi_N(\theta_s^t)$) while the lower-level is updated by $N$ steps of stochastic gradient descent over $\theta_p$.

Therefore, we can get the convergence theorem as in Theorem 3.10 of (Salehi et al., 2025), the BLISS iterates satisfy

$$
\min_{0 \leq t \leq T-1} \mathbb{E} \big[ \|\nabla \Phi(\theta_s^t)\|^2 \big] \leq \frac{C_1}{T\alpha_T} + \frac{C_2}{T\alpha_T} \sum_{t=0}^{T-1} \alpha_t \delta_t^2,
$$

where $C_1, C_2 > 0$ are constants independent of $T$. Following the setting in Corollary 3.13 of Salehi et al. (2025), assuming the error $\delta_t = O(t^{-p})$ with $p \geq 1/4$, we can choose the step size $\alpha_t = O(t^{-q})$ with $q \downarrow 1/2$ (choose $q$ arbitrarily close $1/2$ from above), and then we can achieve the convergence rate of $O(T^{-1/4})$ of the gradient norm.

In practice, the lower-level step budget $N$ is chosen empirically to ensure that the surrogate lower-level solution is sufficiently close to the target solution while keeping the computation affordable. For example, in our 31M proxy setting, we find that 1 lower-level step already gives a reasonably good approximation, whereas in the 160M proxy setting, we use 5 lower-level steps. This suggests that the choice of $N$ depends on the proxy size and is tuned to keep the finite-horizon approximation error sufficiently small in practice. For schedule of learning rate, we apply cosine learning decay to the upper level learning rate $\alpha$.

For choice of the selection ratio, this hyperparameter is decided by the computational budget, since the selected data is used to pretrain the target LLM, which takes over most of the computational resources. This setting also follows form the prior work MATE (Yu et al., 2024) and DsDm (Engstrom et al., 2024), the computational budget limits larger choices of this hyperparameter.

## B. Details of Model Settings

The proxy model and score model serve different purposes: the proxy model acts as a surrogate for the LLM and is trained for next-token prediction, while the score model functions as a regression model that maps individual samples to their corresponding influence scores. To transform the proxy model into the score model, we modify its architecture by replacing the final `Linear` layer with an `AdaptiveAvgPool` layer, followed by a `Linear` layer and a `Sigmoid` activation. Specifically, given the output from the preceding transformer blocks with dimension `[Batch,token_size,Emb_size]`, the `AdaptiveAvgPool` layer computes the average embedding feature across tokens. The `Linear` layer then maps the pooled token representations to a single-dimensional output, which is subsequently passed through a `Sigmoid` activation to produce an influence score within the range $(0, 1)$. In contrast, the proxy model's final `Linear` layer maps features from previous layers to the vocabulary dimension for token prediction.

## C. Implementation Details in LLaMA Experiment

**Model Setup**    In LLaMA setting, the target model is LLaMA-0.5B, and the proxy/score model is LLaMA-134M. They are warmed up under the same process as Pythia setting.

**Training Details of Proxy/Score Model**    There is a little difference in how we deal with the proxy model in LLaMA setting compared to Pythia setting. In addition to resetting the proxy model (LLaMA-134M) at the beginning of each round, we reset it to the initial state every 50 steps of the update of the score model. We distill the target model into the proxy model by minimizing the KL divergence for 240 steps. Then the checkpoint of the proxy model is saved as "initial" state. Since periodically resetting the proxy model ensures a close alignment between two models, we remove the KL divergence regularization term in the lower level loss function. To achieve a better lower-level solution, the proxy model executes 4 lower-level updates, each computed on a batch of 64 samples. After the score model is trained for 50 optimization steps, we reset the proxy model to the initial state.

## D. More Ablation Studies

In this section, we provide more ablation studies to verify the effectiveness of each component in our algorithm design.

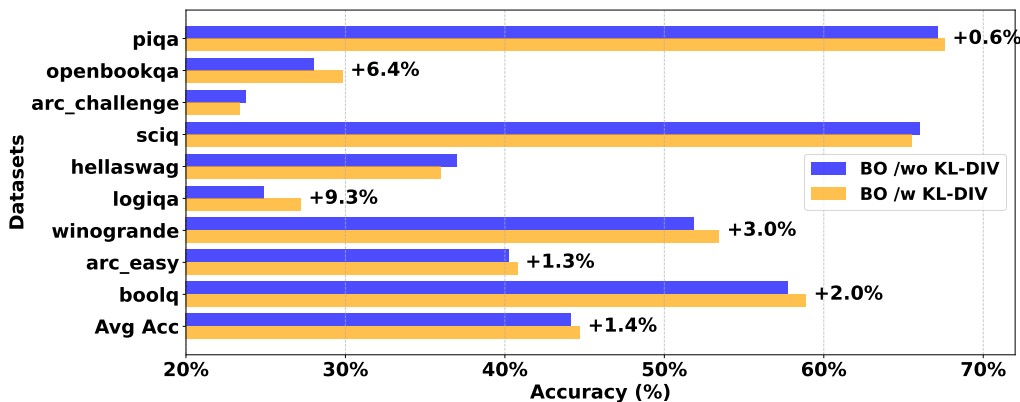

*Figure 3.* The performance comparison of bilevel optimization with/without KL divergence. The number on the bar indicate the accuracy improvement compared to the method without KL divergence.

*Table 5.* Comparison of BLISS with different settings(without softmax and single level update) over multiple downstream datasets (410M model, 10B tokens) with 20k-step training.

| Methods | SciQ | ARC-E | ARC-C | LogiQA | OBQA | BoolQ | HellaSwag | PIQA | WinoGrande | Average |
|---|---|---|---|---|---|---|---|---|---|---|
| Without softmax | 63.5(1.5) | 41.0(1.0) | 22.4(1.2) | 25.7(1.7) | 30.0(2.1) | 52.8(0.9) | 38.8(0.5) | 67.4(1.1) | 51.0(1.4) | 43.6(1.3) |
| Single Level | 64.4(1.5) | 42.3(1.0) | 22.2(1.2) | 24.1(1.7) | 30.6(2.1) | 55.0(0.9) | 39.7(0.5) | 67.1(1.1) | 52.1(1.4) | 44.2(1.3) |
| BLISS | 65.5(1.5) | 40.8(1.0) | 23.4(1.2) | 27.2(1.7) | 29.8(2.0) | 58.9(0.9) | 36.0(0.5) | 67.6(1.1) | 53.4(1.4) | 44.7(1.3) |

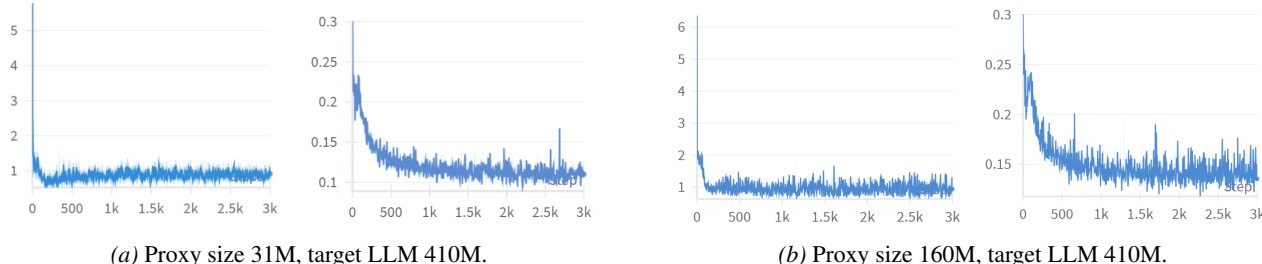

*(a)* Proxy size 31M, target LLM 410M.      *(b)* Proxy size 160M, target LLM 410M.

*Figure 4.* Evolution of lower-level training loss and KL divergence for different proxy model sizes.

*Table 6.* Comparison of BLISS with different size of proxy/score model and on zero-shot evaluation over multiple downstream datasets (410M model, 10B tokens) with 20k-step training.

| Method | SciQ | ARC-E | ARC-C | LogiQA | OBQA | BoolQ | HellaSwag | PIQA | WinoGrande | Average |
|---|---|---|---|---|---|---|---|---|---|---|
| BLISS (Pythia-31M) | 65.5(1.5) | 40.8(1.0) | 23.4(1.2) | 27.2(1.7) | 29.8(2.0) | 58.9(0.9) | 36.0(0.5) | 67.6(1.1) | 53.4(1.4) | 44.7(1.3) |
| BLISS (Pythia-160M) | 63.8(1.5) | 40.8(1.0) | 23.4(1.2) | 27.5(1.8) | 29.8(2.0) | 51.3(0.9) | 38.3(0.5) | 67.6(1.1) | 50.4(1.4) | 44.1(1.3) |

### D.1. Softmax Reparametrization for Score Model's Output

In our experiment, we apply a softmax function on all batch samples' score across GPUs to obtain the importance weights $P_i$. Note that the raw output of the score model is already within the range $(0, 1)$, but we add another softmax function on top of it. We want to demonstrate the effectiveness of this softmax reparameterization. Intuitively, the main benefit is that it naturally amplifies important samples while downweighting less useful ones, improving the overall data selection process.

To assess the impact of the softmax reparameterization, we conduct an ablation experiment comparing two approaches: (i) naive weighting, where the raw outputs of the score model are used directly as sample weights; (ii) softmax weighting, where the softmax-transformed outputs of the score model determine the sample weights. The results, shown in Table 5, indicate that softmax weighting consistently outperforms naive weighting, leading to a 1.1% improvement in average downstream accuracy. This demonstrates that softmax effectively enhances data selection by better distinguishing important samples.

### D.2. The Size of Proxy Model

We conduct experiments using two different sizes of proxy/score models (31M and 160M) for a 410M LLM. We observe that the KL divergence between the proxy and the LLM remains low for both sizes-0.15 for the 160M model and 0.10 for the 31M model. The corresponding learning curves are shown in Figure 4, which presents the results from round 2. The performance comparison of two sizes of proxy model is summarized in Table 6. These findings suggest that even a small proxy model (31M) is sufficient to serve as an effective surrogate for the 410M LLM.

### D.3. Initialization Method for the Score model

In Algorithm 1, we initialize the score model in each new round using the parameters from the last round. This design is motivated by the role of the score model: it learns data representations and ranks the importance of training samples. As training progresses, the model's ability of feature learning improves, making it beneficial to retain learned representations across rounds.

To validate this, we conduct ablation studies comparing two cases:

1. **Original BLISS (BLISS-org)**: the score model in each round is initialized with the parameters from the last round.

2. **Modified Initialization (BLISS†)**: the score model in each round is reset to its initial parameters from round 1.

We then use the trained score models from two cases to select training data and pretrain the target LLM for 15B tokens, respectively. The resulting LLMs are evaluated on multiple downstream datasets. As shown in Table 7, BLISS† achieves an

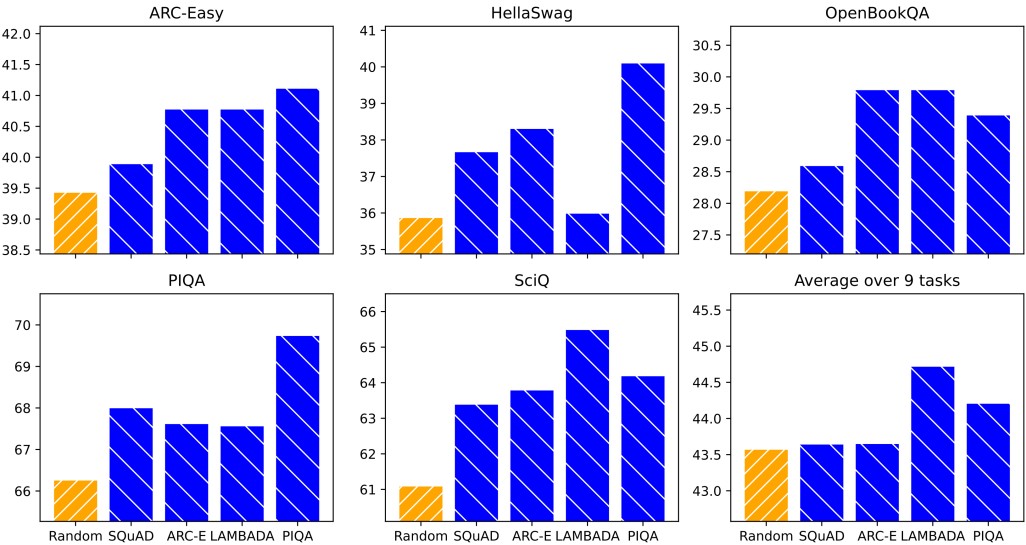

*Figure 5.* Comparison of BLISS trained with different validation datasets (410M model, 10B tokens). We compare our method with different validation datasets with random selection on 1 downstream task in each subplot.

average performance that is $0.4\%$ lower than BLISS-org, demonstrating that continuous initialization leads to better data ranking and improved downstream performance.

*Table 7.* Comparison of methods on zero-shot evaluation over multiple downstream datasets (410M model, 15B tokens). BLISS-org denotes the original algorithm, and BLISS$^\dagger$ is a variant which uses different initialization method for the score model.

| Methods (#FLOPs $\times 10^{19}$) | SciQ | ARC-E | ARC-C | LogiQA | OBQA | BoolQ | HellaSwag | PIQA | WinoGrande | Average |
|---|---|---|---|---|---|---|---|---|---|---|
| BLISS-org | 67.7 (1.5) | 41.7 (1.0) | 23.6 (1.2) | 25.8 (1.7) | 28.4 (2.0) | 56.0 (0.8) | 39.7 (0.5) | 68.7 (1.1) | 53.2 (1.4) | 44.9 (1.3) |
| BLISS$^\dagger$ | 65.2 (1.5) | 41.6 (1.0) | 23.4 (1.2) | 27.1 (1.7) | 29.8 (2.0) | 57.5 (0.8) | 34.9 (0.5) | 67.7 (1.1) | 53.5 (1.4) | 44.5 (1.3) |

### D.4. Validation Datasets

The upper-level optimization aims to minimize the proxy model's loss on the validation dataset, meaning different validation datasets influence data selection. We use different validation set, including SQUAD, ARC-E, LAMBADA, and PIQA, to conduct the bilevel data training, then compare the corresponding downstream performance.

As shown in Figure 5, our algorithm outperforms random selection on most downstream tasks, except BoolQ, regardless of the validation dataset. Notably, LAMBADA yields the highest average accuracy, improving 1.15% over random selection, likely due to its broad domain coverage.

We additionally experimented with a mixed validation set formed by combining multiple validation datasets (SQuAD, ARC-E, LAMBADAM, PIQA), and obtained an average downstream accuracy of 44.08% after 10B-token pretraining. When compared with the *Average* result (the last subfigure) in Figure 5, this result is lower than the best single-validation-set choice, where LAMBADA achieves the highest average accuracy among all tested validation sets. Therefore, our current evidence suggests that simply mixing multiple validation sets does not necessarily improve the upper-level supervision signal for BLISS. A possible reason is that LAMBADA used in (Engstrom et al., 2024; Yu et al., 2024) provides a more coherent notion or signal of downstream utility for data selection.

We also notice that our averaged performance is greatly affected by the accuracy of BoolQ task across all validation datasets. This indicates that it is hard to learn when the answer is too short like yes or no.

# E. Additional results

Since we use the same experimental settings as MATES(Yu et al., 2024), including pretraining model, data and training steps, we evaluate MATES on the downstream tasks with their checkpoint model (`https://huggingface.co/yuzc19/pythia-410m-mates/blob/main/iter-200800-ckpt.pth`) of 50k steps. For other baselines, we quote Table 1 from MATES(Yu et al., 2024) for convenience of look-up for the performance of more algorithms.

*Table 8.* Results of Different Methods under the 410M/1B Setting. Subscripts denote standard error. Best scores are in bold.

| Methods($\#FLOPs \times 10^{19}$) | SciQ | ARC-E | ARC-C | LogiQA | OBQA | BoolQ | HellaSwag | PIQA | WinoGrande | Average |
|---|---|---|---|---|---|---|---|---|---|---|
| **410M Setting:** 410M model, 25B tokens | | | | | | | | | | |
| Random(6.35) | $64.1_{(1.5)}$ | $40.2_{(1.0)}$ | $\mathbf{25.6}_{(1.3)}$ | $24.7_{(1.7)}$ | $29.4_{(2.0)}$ | $58.9_{(0.9)}$ | $39.7_{(0.5)}$ | $67.1_{(1.1)}$ | $50.6_{(1.4)}$ | $44.5_{(1.3)}$ |
| DSIR(6.35) | $63.1_{(1.5)}$ | $39.9_{(1.0)}$ | $23.8_{(1.2)}$ | $27.0_{(1.7)}$ | $28.4_{(2.0)}$ | $58.3_{(0.9)}$ | $39.6_{(0.5)}$ | $66.8_{(1.1)}$ | $51.5_{(1.4)}$ | $44.3_{(1.3)}$ |
| LESS(246.35) | $64.6_{(1.5)}$ | $42.3_{(1.0)}$ | $23.1_{(1.2)}$ | $25.2_{(1.7)}$ | $30.4_{(2.1)}$ | $55.6_{(0.9)}$ | $\mathbf{41.9}_{(0.5)}$ | $67.2_{(1.1)}$ | $51.0_{(1.4)}$ | $44.6_{(1.4)}$ |
| SemDeDup(7.81) | $63.5_{(1.5)}$ | $\mathbf{42.4}_{(1.0)}$ | $24.4_{(1.3)}$ | $\mathbf{27.6}_{(1.7)}$ | $30.0_{(2.1)}$ | $58.2_{(0.9)}$ | $40.8_{(0.5)}$ | $67.8_{(1.1)}$ | $52.3_{(1.4)}$ | $45.2_{(1.3)}$ |
| DsDm(10.72) | $65.4_{(1.5)}$ | $41.7_{(1.0)}$ | $24.7_{(1.3)}$ | $27.5_{(1.8)}$ | $29.0_{(2.1)}$ | $57.5_{(0.9)}$ | $40.3_{(0.5)}$ | $67.1_{(1.1)}$ | $50.1_{(1.4)}$ | $44.9_{(1.4)}$ |
| QuRating(26.35) | $64.8_{(1.5)}$ | $42.0_{(1.0)}$ | $25.4_{(1.3)}$ | $25.3_{(1.7)}$ | $30.2_{(2.1)}$ | $58.9_{(0.9)}$ | $40.7_{(0.5)}$ | $67.5_{(1.1)}$ | $52.1_{(1.4)}$ | $45.2_{(1.4)}$ |
| MATES(8.11) | $65.7_{(1.5)}$ | $41.5_{(1.0)}$ | $25.0_{(1.3)}$ | $26.1_{(1.7)}$ | $\mathbf{30.8}_{(2.1)}$ | $\mathbf{60.6}_{(0.9)}$ | $41.0_{(0.5)}$ | $67.8_{(1.1)}$ | $51.8_{(1.4)}$ | $45.7_{(1.4)}$ |
| BLISS(8.08) | $\mathbf{68.1}_{(1.5)}$ | $42.2_{(1.0)}$ | $25.1_{(1.3)}$ | $27.3_{(1.7)}$ | $29.6_{(2.0)}$ | $59.3_{(0.9)}$ | $41.2_{(0.5)}$ | $\mathbf{68.2}_{(1.1)}$ | $\mathbf{52.0}_{(1.4)}$ | $\mathbf{45.9}_{(1.4)}$ |
| **1B Setting:** 1B model, 25B tokens | | | | | | | | | | |
| Random(17.67) | $65.8_{(1.5)}$ | $43.7_{(1.0)}$ | $25.6_{(1.3)}$ | $27.5_{(1.8)}$ | $31.8_{(2.1)}$ | $60.2_{(0.9)}$ | $43.8_{(0.5)}$ | $68.9_{(1.1)}$ | $50.7_{(1.4)}$ | $46.4_{(1.4)}$ |
| DSIR(17.67) | $65.8_{(1.5)}$ | $42.6_{(1.0)}$ | $24.7_{(1.3)}$ | $\mathbf{28.7}_{(1.8)}$ | $29.2_{(2.0)}$ | $59.7_{(0.9)}$ | $44.2_{(0.5)}$ | $68.3_{(1.1)}$ | $\mathbf{53.2}_{(1.4)}$ | $46.3_{(1.4)}$ |
| SemDeDup(19.13) | $66.8_{(1.5)}$ | $45.5_{(1.0)}$ | $25.3_{(1.3)}$ | $27.6_{(1.8)}$ | $30.6_{(2.1)}$ | $60.2_{(0.9)}$ | $45.3_{(0.5)}$ | $69.7_{(1.1)}$ | $52.5_{(1.4)}$ | $47.1_{(1.4)}$ |
| DsDm(22.04) | $68.2_{(1.5)}$ | $45.0_{(1.0)}$ | $\mathbf{26.5}_{(1.3)}$ | $26.6_{(1.7)}$ | $29.4_{(2.0)}$ | $59.0_{(0.9)}$ | $44.8_{(0.5)}$ | $68.9_{(1.1)}$ | $51.9_{(1.4)}$ | $46.7_{(1.3)}$ |
| QuRating(37.67) | $67.1_{(1.5)}$ | $45.5_{(1.0)}$ | $25.6_{(1.3)}$ | $26.9_{(1.7)}$ | $29.8_{(2.0)}$ | $60.3_{(0.9)}$ | $45.2_{(0.5)}$ | $70.2_{(1.1)}$ | $51.6_{(1.4)}$ | $46.9_{(1.3)}$ |
| MATES(19.97) | $67.3_{(1.5)}$ | $44.9_{(1.0)}$ | $25.9_{(1.3)}$ | $\mathbf{28.7}_{(1.8)}$ | $32.2_{(2.1)}$ | $\mathbf{60.9}_{(0.9)}$ | $45.3_{(0.5)}$ | $69.5_{(1.1)}$ | $52.4_{(1.4)}$ | $47.5_{(1.4)}$ |
| BLISS(19.53) | $\mathbf{69.4}_{(1.5)}$ | $\mathbf{45.7}_{(1.0)}$ | $24.8_{(1.3)}$ | $25.8_{(1.7)}$ | $\mathbf{33.2}_{(2.1)}$ | $59.8_{(0.9)}$ | $\mathbf{47.8}_{(0.5)}$ | $\mathbf{71.6}_{(1.1)}$ | $52.9_{(1.4)}$ | $\mathbf{47.9}_{(1.3)}$ |

# F. Experimental Hyperparameters

*Table 9.* Experimental settings.

| Hyperparameters | Values |
|---|---|
| *Pretrain* | |
| Data set | C4 |
| Tokens | 25B |
| Model | Pythia-410M/1B/2.8B, LLaMA-0.5B |
| batch size | 512 |
| Sequence length | 1024 |
| Max learning rate | 1e-3 |
| *bilevel optimization* | |
| Proxy/Score model | Pythia-31M (for 410M LLM), Pythia-160M (for 1B LLM), LLaMA-134M (for LLaMA-0.5B LLM) |
| $\gamma$ | 1e-2 |
| $\lambda$ | 1e-6 |
| batch size | 16(Pythia-410M)/32(Pythia 1B) |
| Proxy/Score model learning rate($\eta_1/\alpha$) | 1e-5 |
| GDLS learning rate($\eta_2$) | 1e-2 |
| GDLS steps($K$) | 3 |
| Score model steps | 3k(Pythia-410M/1B)/1k(LLaMA-0.5B) |
| Proxy model steps | 3k(Pythia-410M/1B)/1k(LLaMA-0.5B) |
| Initialization of score/proxy model | Randomly initialized |

Table 9 shows the hyperparameter settings in our experiments. We use cosine learning rate scheduler in bilevel optimization, WSD(Yu et al., 2024) learning rate scheduler for pretraining and constant learning rate for GDLS. We use double loop to update the proxy model when employing 1B LLM, i.e., 5 steps for the lower level update. The experiments run on 8 A6000 GPUs with Distributed Data Parallel (DDP) strategy.

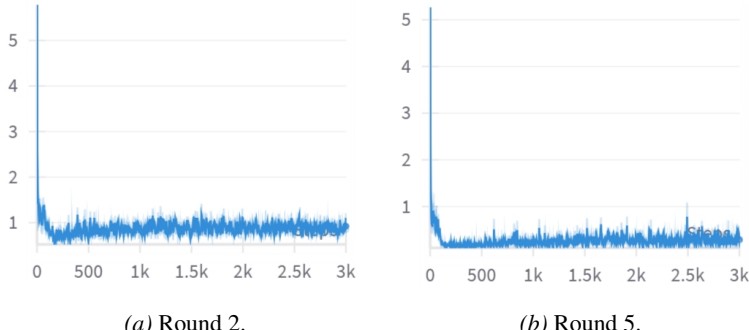

*(a)* Round 2.         *(b)* Round 5.

*Figure 6.* Evolution of training loss within a bilevel training round.

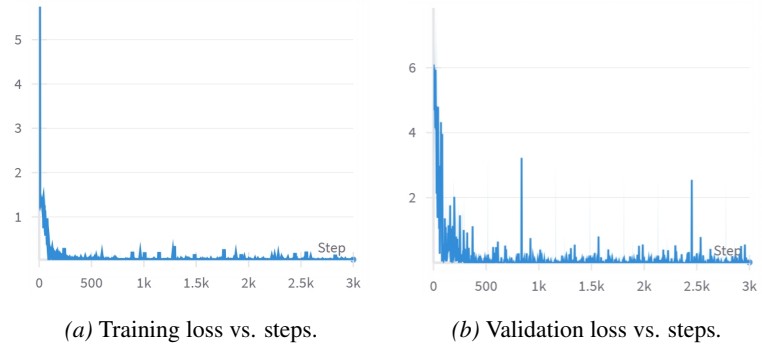

*(a)* Training loss vs. steps.        *(b)* Validation loss vs. steps.

*Figure 7.* Lower-level training loss and upper-level validation loss in round 2 of bilevel optimization without KL divergence.

## G. Evolution of training and validation loss

In Figure 6a, 6b, we visualize the curves training loss pretraining round 2 and 5.

## H. Distributed Softmax to Compute Influence Score

In bilevel optimization, the importance weight $P_i$ is computed based on a mini batch that is distributed across different GPUs. However, back propagation through different GPUs is not implemented by Pytorch. Thus we deploy "distributed softmax" in practice. In detail, our implementation requires 3 times of communication among GPUs.

$$P_i = \frac{e^{h(\theta_s;\xi_i)}}{\sum_{j=1}^{B} e^{h(\theta_s;\xi_j)}}. \tag{7}$$

As equation (7) shows, the denominator of $P_i$ is the summation of every sample's exponential score. Therefore, in the first communication, each GPU gets the scores from others and calculates the denominator locally. A second communication is required to compute the term $\sum_{j=1}^{\mathcal{B}} P_j \nabla_{\theta_s} h(\theta_s^t; \xi_j)$ in equation (5). In detail, we need to gather gradients of $h$ and $\mathcal{L}$' of every sample across all GPUs. After computing hyper-gradients of every sample, they are accumulated to update upper-level variables. With efficient communication API provided by Fabric `https://lightning.ai/docs/fabric/stable/`, the time consumed in bilevel optimization of each round is within 1.5 hours.

## I. Running Time and Memory

We measured the memory and runtime of the data selection stage for both BLISS and MATES under different target (or pretraining) model sizes (for short, T: target). The results are shown in Table 10. We have two observations:

- BLISS scales well with larger target models. Note that the target model is not an optimization variable for the bilevel optimization and it is only used for calculating the KL divergence. Therefore, it does not affect the scalability of

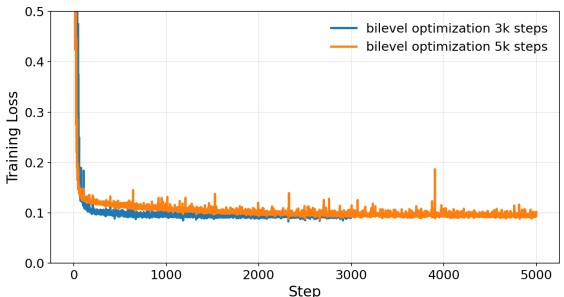

*Figure 8.* Training loss with different steps.

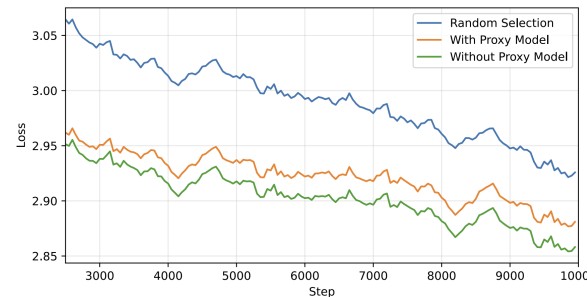

*Figure 9.* Test loss on SlimPajama-6B.

bilevel optimization. When increasing the target from 410M to 1B, BLISS's memory and runtime grow moderately ($49.46 \rightarrow 74.51$ GB; $5.03 \rightarrow 11.82$ hours), as expected.

- BLISS is significantly faster than MATES. MATES incurs high cost because each round requires oracle data collection. For every example, MATES performs a one-step gradient update on the target model and evaluates the validation loss change to compute influence scores. This per-example simulation dominates runtime. In contrast, BLISS avoids all per-example oracle evaluations in MATES and performs bilevel optimization directly on the proxy/score model, leading to $2 - 3\times$ faster data selection.

*Table 10.* The comparison of runtime and memory in data selection stage.

| Setting | Memory peak (GB) | Total data selection time (hours) |
|---|---|---|
| MATES: T: 410M | 36.36 | 18.0 |
| MATES: T: 1B | 63.52 | 30.32 |
| BLISS: T: 410M | 49.46 | 5.03 |
| BLISS: T: 1B | 74.51 | 11.82 |

## J. Ablation Study for Bilevel Optimization Steps

We did the ablation study to investigate the steps of bilevel optimization on Pythia models. As shown in Figure 8, both the 3k-step and 5k-step settings converge to nearly the same training loss. This indicates that 3k steps are sufficient for the Pythia-type proxy model, as increasing the steps to 5k does not yield additional improvements. So we fix the training steps of bilevel optimization to 3k steps in main experiments.

## K. Domain Reweighting: Fidelity of Proxy Models

To verify the fidelity of proxy models to full-scale LLMs, we conduct a domain-reweighting experiment on the SlimPajama-6B dataset (DKYoon, 2023), which is a multi-domain dataset, including ArXiv, Books, C4, CommonCrawl, GitHub, StackExchange, and Wikipedia. The objective is to learn optimal domain weights $\alpha \in \mathbb{R}^d$, $d$ is the domain size, such that a model trained on data sampled according to the weights achieves the best downstream performance.

We compare two settings:

1. **Case 1 (with proxy model):** The lower level optimizes a lightweight proxy model (LLaMA-134M) with output alignment to the target LLM (LLaMA-300M), and the upper level learns the domain weights $\tilde{\alpha}$.

2. **Case 2 (without proxy model):** The lower level directly optimizes the target LLM (LLaMA-300M), and the upper level learns the domain weights $\alpha$.

We perform bilevel optimization for 1,000 steps in both cases to learn the domain weights, where $10\%$ of the original training set is held out as the validation set for the upper-level objective, and the remaining $90\%$ is used as the lower-level

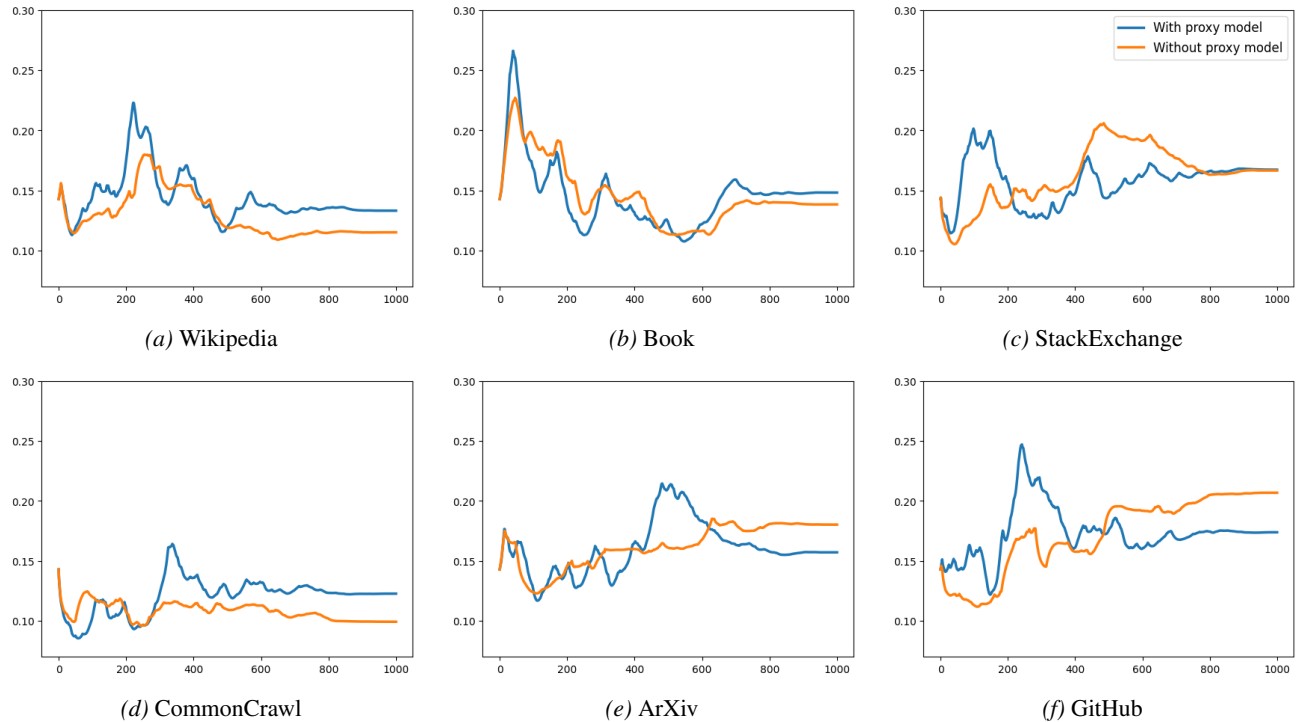

*Figure 10.* Learning curves of domain weights.

training set. After obtaining $\tilde{\alpha}$ and $\alpha$, we train two final LLaMA-300M models on data sampled according to each set of weights, respectively . Figure 10 presents the learning curves of domain weights for both cases. We observe that the trajectories of $\tilde{\alpha}$ and $\alpha$ are highly similar across most domains (e.g., in the domain of Wikipedia, Book, Stackexchange), demonstrating that the proxy model maintains high fidelity to the full-scale LLM in data selection.

Finally, we evaluate the resulting pretrained LLMs on the test set of SlimPajama-6B, and the results are shown in Figure 9. The test loss curves show that data selection based on the proxy model maintains high fidelity to the full-scale LLM, while significantly outperforming random selection.

