# OpenReview forum: "BLISS: A Lightweight Bilevel Influence Scoring Method for Data Selection in Language Model Pretraining"
_ICML.cc/2026/Conference — ICML 2026 regular_

### Official Review · Reviewer_6Mm2 · 2026-02-24

**Soundness:** 2
**Presentation:** 3
**Significance:** 2
**Originality:** 2
**Overall Recommendation:** 4
**Confidence:** 2

**Summary:**

The paper proposed the BileveL Influence Scoring method for data Selection (BLISS) to select the pretraining data for LLMs. It addresses two problems of the existing methods: reliance on the external pretrained model and neglect of the long-term effect. The authors model the dataset selection as a bilevel optimization problem, where the upper-level objective optimizes the score model to assign importance weights to training samples, and the lower-level objective is to optimize a proxy model that behaves similarly to the LLM by the scored samples.

**Compliance With Llm Reviewing Policy:**

Affirmed.

**Final Justification:**

Thank you for the detailed rebuttal. The additional empirical evidence helps clarify the validity of the proposed bilevel optimization framework, and I find the convergence analysis and efficiency improvements to be well-supported.
While I still hold a somewhat conservative view regarding the formulation of the bilevel objective—specifically, it appears more heuristically motivated than analytically grounded—the rebuttal partially alleviates my concerns about its practical effectiveness.

**Key Questions For Authors:**

### Theory
- Can you provide a formal justification for why the lower-level objective should produce a proxy model whose training dynamics are predictive of (or aligned with) the target LLM’s dynamics? Concretely, are there assumptions or conditions under which the bilevel procedure provably improves data selection (e.g., convergence to a meaningful solution, stability guarantees, or generalization guarantees)?

### Experiments
- How robust are the gains beyond the single training dataset used in the paper?

**Limitations:**

yes

**Strengths And Weaknesses:**

## Strengths

The paper clearly motivates its contributions by identifying concrete limitations in prior work and formulating a principled solution via a bilevel optimization framework. The proposed method is well-supported empirically, showing consistent improvements over strong state-of-the-art baselines while also reducing pre-training time. In addition, the experimental section is particularly thorough: it reports both performance and runtime, and includes component-wise analyses that clarify the contribution and practical effect of each part of the bilevel formulation.

## Weaknesses

### Soundness

- The motivation for the bilevel formulation is not sufficiently clarified. In particular, it is unclear why the proposed lower-level objective would cause the proxy model to behave in a way that reliably reflects or predicts the behavior of the target LLM.
- If the proxy model is intended to guide LLM pretraining, it should ideally (i) start from behavior that is reasonably aligned with the LLM and (ii) evolve consistently as the LLM is updated. The paper does not clearly explain why the lower-level optimization design guarantees or even encourages these properties.
- The method lacks theoretical guarantees, such as convergence results, generalization bounds, or formal conditions under which the bilevel procedure provably improves data selection or downstream performance.

### Presentation

- The intuition behind the bilevel design is not clearly articulated. The paper would benefit from a more explicit explanation of what each level optimizes, how the two levels interact, and why this interaction addresses the limitations of prior work.

### Significance

- The empirical evaluation is limited to a single training dataset, which makes it difficult to assess the robustness and generalizability of the results.
- The reported accuracy improvements appear modest, and it is unclear whether they are practically significant across different training datasets.

### Originality

- Although applying bilevel optimization to dataset selection is a reasonable direction, the paper does not clearly explain why this specific bilevel construction is particularly well-suited to the problem.
- The connection between the proposed formulation and improved efficiency or effectiveness in data selection is asserted but not convincingly demonstrated.

---

> ### Author Rebuttal · Authors · 2026-03-31
>
> **Q1. It is unclear why the proposed lower-level objective would cause the proxy model to behave in a way that reliably reflects or predicts the behavior of the target LLM.**\
> **A1.** The lower-level objective is designed so that the proxy does not act as an independent small LM, but instead tracks the target LLM's data preference through weighted training together with KL alignment. We provide direct empirical evidence for this point in **Appendix J**. In a domain-reweighting experiment on SlimPajama-6B, we compare a proxy-based bilevel setting (LLaMA-134M aligned to LLaMA-300M) against a bilevel setting that directly uses the full target LLM in the lower level. The learned domain-weight trajectories are highly similar across most domains (6/7) in Figure 10, directly showing that the proxy preserves the target model's preference for data selection.
>
> Section 6.2 also shows that the KL regularization term helps the behavior alignment between proxy and target LLM. Removing this term will hurt downstream selection quality.
>
> **Q2.  The paper does not clearly explain why the lower-level optimization design guarantees or even encourages these properties.**\
> **A2.** Our lower-level design is intended to encourage exactly these two properties. First, the lower-level objective contains a KL-divergence term between the proxy model and the target LLM on the softmax-normalized output logits. Therefore, to optimize the lower level well, the proxy must keep this KL term small, which explicitly encourages the proxy to stay close to the predictive behavior of the target LLM rather than drifting to an arbitrary surrogate. Second, this alignment is enforced throughout the bilevel optimization process, not only at initialization: as the target LLM is updated round by round, the proxy is repeatedly optimized under the weighted training loss together with the KL term, so the proxy is encouraged to evolve consistently with the current state of the target model. In this sense, the KL term is exactly the mechanism that keeps the proxy aligned with the target LLM across training, making the learned sample preferences more likely to remain effective for downstream data selection. Figure 4 in Appendix C.2 shows the small value of KL divergence after 500 lower level training steps.
>
> **Q3. The method lacks theoretical guarantees**\
> **A3.** We have provided a theoretical analysis of convergence, please see A1 in our response to Reviewer bj47 for more details.
>
> **Q4. The intuition behind the bilevel design is not clearly articulated.**\
> **A4.** Intuitively, the lower level asks: *given a candidate weighting over training samples, what proxy model will be obtained after training?* It trains the proxy on the weighted training loss, together with a KL term that aligns the proxy with the target LLM. The upper level then asks: *are these weights useful for downstream generalization?* It updates the score model so that the resulting proxy achieves better validation performance. Hence, the two levels interact naturally: the score model proposes sample weights, the proxy model reflects their dynamic training effect, and the validation loss improves the score model. This design addresses prior limitations by capturing multi-step impact beyond one-step influence methods and by avoiding reliance on external pretrained oracle models.
>
> **Q5. The empirical evaluation is limited to a single training dataset.**\
> **A5.** We would like to clarify that we do provide additional evidence beyond C4 in *Appendix J*. Specifically, we conduct a *Domain Reweighting: Fidelity of Proxy Models* experiment on a different multi-domain dataset, SlimPajama-6B, and compare proxy-based bilevel optimization with full-model bilevel optimization. The learned domain-weight trajectories are highly consistent with target LLM across most domains (6/7), which supports that the proxy preserves the target model's data preference on a different corpus.
>
> **Q6. The improved efficiency or effectiveness in data selection is not convincingly demonstrated.**\
> **A6.** BLISS is more efficient than prior work like MATES because it avoids costly oracle-based per-sample influence estimation. Specifically, MATES evaluates each training sample by measuring the loss change after a one-step gradient update, which requires per-sample gradients and prevents effective scaling of batch size on each GPU, making the selection process highly expensive. In contrast, BLISS casts data selection as a bilevel optimization problem and amortizes the selection signal through a lightweight score model and proxy model, which can be optimized to convergence in relatively few steps (e.g., 3,000 per round). This makes BLISS substantially more computationally efficient. For example in 1B setting in Table 4, MATES consumes $2.30 \times 10^{19}$   (0.83+0.01+1.46) FLOPS for data selection, in contrast, BLISS consumes $1.86 \times 10^{19}$ (0.07+0.26+1.53), which reduces the computational cost by 19.1\%.

---

> > ### Author Rebuttal · Reviewer_6Mm2 · 2026-04-02
> >
> > Thank you for the thorough rebuttal. The discussion on convergence, proxy alignment, and computational efficiency was helpful, and these points are now much clearer to me.
> > My remaining reservation is mainly about the formulation itself. Although the bilevel design is intuitive and supported by experiments, it still feels somewhat heuristic, and I would have liked a more principled justification for why this particular objective is the right one. I also find the empirical gains somewhat limited.
> > As a result, I view the work positively overall, but at the level of a weak accept.

---

> > > ### Author Response · Authors · 2026-04-05
> > >
> > > We sincerely appreciate your valuable feedback and thoughtful review. We would also be happy to address any further questions you may have.

---

### Official Review · Reviewer_bj47 · 2026-03-08

**Soundness:** 2
**Presentation:** 3
**Significance:** 2
**Originality:** 3
**Overall Recommendation:** 4
**Confidence:** 4

**Summary:**

This paper proposes BLISS (BileveL Influence Scoring method for data Selection), a framework for selecting high-quality data for Large Language Model (LLM) pretraining. Unlike existing approaches that rely on expensive external pretrained “oracle” models or heuristic filtering, BLISS operates entirely from scratch using a bilevel optimization formulation. Specifically, BLISS employs a small proxy model and a score model. The score model learns sample-wise importance weights by solving a bilevel optimization problem: at the lower level, the proxy model is trained to convergence on the weighted training data; at the upper level, the weights are adjusted to minimize the proxy model’s loss on a validation set.The authors further introduce a KL-divergence regularization term to encourage alignment between the proxy model and the target LLM. Experiments on Pythia (410M, 1B, 2.8B) and LLaMA architectures demonstrate that BLISS outperforms state-of-the-art methods such as MATES and random selection on downstream tasks.

**Compliance With Llm Reviewing Policy:**

Affirmed.

**Final Justification:**

The paper proposes a novel bilevel framework for pretraining data selection without relying on external pretrained oracles.

I still have some reservations about the strength of the "long-term effect claim. The rebuttal usefully reframes the method as optimizing a finite-horizon surrogate induced by K-step proxy updates, but in practice K remains quite small, so it is still unclear how well this supports the claimed long-term influence. In addition, resetting the proxy at each round seems to weaken the interpretation that the method captures persistent cross-round effects.

That said, I find the method technically interesting and the empirical study reasonably strong. On balance, I view these issues as limitations in justification and claim calibration rather than fatal flaws, so I am updating my score to weak accept.

**Key Questions For Authors:**

1. Can the authors provide theoretical justification or empirical evidence demonstrating robustness to variations in key choices?
2. How sensitive is performance to the specific choice of validation dataset? Would aggregating multiple benchmarks perform better?

**Limitations:**

yes

**Strengths And Weaknesses:**

Strengths
1. The authors address a limitation in current data selection research: the reliance on pretrained models to evaluate data quality. BLISS eliminates this dependency by training from scratch with a lightweight proxy model, making the framework self-contained.
2. By formulating data selection as a bilevel optimization problem in which the lower-level model is trained for a longer horizon, BLISS aims to more accurately capture the long-term impact of training samples over the course of optimization.
3. The paper conducts extensive experiments across different model sizes and architectures.

Weakness
1. Several critical experimental settings appear largely empirical, with limited theoretical justification. For example, the selection ratio, the number of proxy model update steps, and the interval for resetting the proxy model. The paper does not provide sufficient analysis explaining why these specific configurations work.
2. Although BLISS improves computational efficiency, its performance gains over strong baselines are often modest. For example, in Table 1 (410M setting), the average performance differs by only 0.2% (45.7% vs. 45.9%).
3.The authors primarily use LAMBADA as the validation dataset, but it is unclear why they did not construct a more comprehensive validation set by aggregating multiple evaluation benchmarks.
4. As acknowledged in Section 5.5 and Table 10, computing Hessian-vector products (HVPs) for bilevel optimization increases peak memory consumption relative to alternative methods. This may limit scalability.

---

> ### Author Rebuttal · Authors · 2026-03-31
>
> **Q1. Limited theoretical justification for critical experimental settings.**\
> **A1.** We thank the reviewer for this question. Classical bilevel convergence analyses typically rely on a strongly-convex lower-level (LL) problem, so that the lower-level solution mapping is relatively easy to solve. This assumption also underlies many recent single-loop bilevel methods which update the upper-level (UL) and lower-level variables with the same frequency [SOBA](https://arxiv.org/pdf/2201.13409). However, this formulation is not suitable for BLISS. In our setting, the lower-level corresponds to training a proxy language model under score-induced sample weights, which is a nonconvex optimization problem rather than a strongly-convex problem. Therefore, the classical single-loop bilevel optimization algorithm is not applicable here since the exact minimizer for the lower-level problem is not easy to solve due to nonconvexity.
>
> To address this issue, our proposed method instead adopts a finite-horizon view and redefines the lower level through the multi-step stochastic gradient descent (SGD) on the proxy model for $K$ steps.
> $$
> \min\_{\theta\_s}\hat{\Phi}(\theta\_s):=\mathbb{E}\_{\zeta\sim\mathcal D\_{\mathrm{val}}}[F(\theta\_p^{K}(\theta\_s);\zeta)] \quad \text{(UL)}
> \quad\text{s.t.}\quad
> \theta\_p^{K}(\theta\_s)=\mathrm{SGD}(G(\theta\_p,\theta\_s;\xi)),\ \xi\sim\mathcal D\_{\mathrm{tr}} \quad \text{(LL)}
> $$
> Concretely, given $\theta_s$, the proxy model is trained for $K$-step SGD, and the upper-level objective is defined on the surrogate of optimal proxy model. This leads to a finite-horizon surrogate objective $\hat{\Phi}(\theta_s)$ that is more faithful to our algorithm, since in each upper iteration the score model interacts with the proxy obtained from a fixed training budget, not with an idealized lower-level optimum.
>
> Assume that the $K$-step proxy update map is Lipschitz in $\theta_s$, the validation objective is smooth and lower bounded, and the stochastic hypergradient estimator for the upper-level update has bounded bias and bounded second moment. Then, after $T$ outer iterations, the iterates of BLISS satisfy $
> \frac{1}{T}\sum_{t=0}^{T-1}\mathbb{E}\\|\nabla \Phi(\theta_s^t)\\|
> \le
> \delta_K + \frac{1}{T}\sum_{t=0}^{T-1}\mathbb{E}\\|\nabla \hat\Phi(\theta_s^t)\\|,
> $ where $
> \delta_K
> :=
> \sup_{\theta_s}\\|\nabla \Phi(\theta_s)-\nabla \hat\Phi(\theta_s)\\|
> $
> denotes the approximation error induced by replacing the ideal lower-level solution with the $K$-step proxy training dynamics, which goes to zero when $K$ goes to infinity. The exact rate of $\delta_K$ would depend on the smoothness conditions of $F$ and $G$ and we expect it to be upper bounded by $O(\text{poly}(1/K))$ (e.g., Theorem 2.1 of [this paper](https://arxiv.org/pdf/2511.06774)). The second term is the estimated gradient norm of upper-level problem and can be upper bounded by $\mathcal{O}(T^{-1/4})$ by standard analysis of first-order methods.
>
> In practice, the lower-level step budget $K$ is chosen empirically to ensure that the surrogate lower-level solution is sufficiently close to the target solution while keeping the computation affordable. For example, in our 31M proxy setting, we find that 1 lower-level step already gives a reasonably good approximation, whereas in the 160M proxy setting, we use 5 lower-level steps. This suggests that the choice of $K$ depends on the proxy size and is tuned to keep the finite-horizon approximation error sufficiently small.
>
> For choice of the selection ratio, this hyperparameter is decided by the computational budget, since the selected data is used to pretrain the target LLM, which takes over most of the computational resources. This setting also follows form the prior work [MATES](https://arxiv.org/pdf/2406.06046) and [DsDm](https://arxiv.org/pdf/2401.12926), the computational budget limits larger choices of this hyperparameter.
>
> BLISS operates in rounds, and Eq. (1) describes data selection in a single round. In practice, however, the selected data distribution can change substantially across rounds, so $\xi$ may come from a different distribution, leading to a different lower-level problem. As a result, warm-starting from the previous proxy can introduce bias toward the previously selected distribution and reduce the proxy's ability to adapt to the current round, i.e., its *plasticity*. This intuition is consistent with prior work [Dash](https://arxiv.org/pdf/2410.23495), which suggests that under non-stationary training distributions, a long-trained network can become harder to adapt than a freshly initialized one. Therefore, we reset the proxy $\theta_p$ at each round so it can better fit the newly training distribution and maintain alignment with the target LLM, rather than inheriting optimization bias from earlier rounds.
>
> **Q2. What is the result on a comprehensive validation set.**\
> **A2.** We have completed the experiment and please see A4 in our response to Reviewer 2DtU for more details.

---

> > ### Author Rebuttal · Reviewer_bj47 · 2026-04-04
> >
> > The paper proposes a novel bilevel framework for pretraining data selection without relying on external pretrained oracles.
> >
> > I still have some reservations about the strength of the "long-term effect claim. The rebuttal usefully reframes the method as optimizing a finite-horizon surrogate induced by K-step proxy updates, but in practice K remains quite small, so it is still unclear how well this supports the claimed long-term influence. In addition, resetting the proxy at each round seems to weaken the interpretation that the method captures persistent cross-round effects.
> >
> > That said, I find the method technically interesting and the empirical study reasonably strong. On balance, I view these issues as limitations in justification and claim calibration rather than fatal flaws, so I am updating my score to weak accept.

---

> > > ### Author Response · Authors · 2026-04-05
> > >
> > > We sincerely appreciate your valuable feedback and thoughtful review. You raise a very fair point regarding our use of the term "long-term effect'', particularly given our experimental setting where the lower-level horizon is relatively small (e.g., $K=5$ in our 1B setting experiment). We agree that this phrasing could lead to misunderstandings regarding the scope of our claims and the interpretation of our cross-round effects.
> > >
> > > To address this, we will replace the phrase "long-term effect" with the more precise term "dynamic data preference'' throughout the revised manuscript. Our intended point is that BLISS captures the dynamic data preferences of the target LLM under continuous model updates, rather than claiming to model an arbitrarily long optimization horizon. Existing methods such as [MATES](https://arxiv.org/pdf/2406.06046) compute the sample influence using the fixed current target LLM, and thus cannot dynamically capture the data impact on performance of the target model during model updates. Instead, BLISS, optimized through bilevel updates, ensures that the proxy model is dynamically driven toward an approximately optimal solution given an arbitrary training mini-batch at the current stage. Consequently, the score model does not output fixed sample scores; instead, it continuously adjusts sample importance as the proxy model is continuously updated. Furthermore, the KL term explicitly aligns the proxy with the target LLM, ensuring these learned preferences remain highly informative for the target model.
> > >
> > > Finally, regarding the proxy reset, we want to clarify that reinitializing the proxy at each round is an experimental choice for for consistency with the baseline setup, not a methodological limitation. In our experiments, we adopted a round-based data selection setup consistent with prior work such as [MATES](https://arxiv.org/pdf/2406.06046). Under this setup, resetting the proxy at the beginning of each round provides a clean surrogate for the current round’s data distribution and target-model state. If the data were provided in a single, continuous round, we would not need to reinitialize the proxy model. We will update the text to explicitly state that this reset was driven by the baseline setup rather than a fundamental necessity for BLISS.
> > >
> > > We would be happy to clarify any further questions that may help address your concerns. Thank you again for your time and thoughtful review.

---

### Official Review · Reviewer_BzUh · 2026-03-13

**Soundness:** 3
**Presentation:** 3
**Significance:** 3
**Originality:** 3
**Overall Recommendation:** 4
**Confidence:** 3

**Summary:**

The paper introduces BLISS, a data selection method for model pre-training that leverages a small proxy model and a scoring model to estimate the long-term influence of each training sample. The authors further formulate the data selection problem as a bilevel optimization task for jointly training the proxy model and the scoring model. Experimental results show that the proposed method achieves ~1.7x training speedups when training on the selected data subset.

**Compliance With Llm Reviewing Policy:**

Affirmed.

**Key Questions For Authors:**

See weakness

**Limitations:**

Yes

**Strengths And Weaknesses:**

## Strengths
1. The paper proposes BLISS, a data selection method for model pre-training from scratch.
2. The paper formulates the data selection problem as a bilevel optimization task.
3. Experimental results demonstrate the effectiveness of the proposed method.


## Weakness
1. The paper does not provide a detailed discussion of the influence scoring mechanism. It is unclear how the proposed approach differs from prior methods on influence estimation such as [1, 2].
2. Although the method is claimed to target LLM pre-training, the models used in the experiments appear relatively small (2.8B).
3. The paper lacks comparisons with some recent baselines, such as Alinfik [3].


[1] Lin, Huawei, et al. "Token-wise influential training data retrieval for large language models." Proceedings of the 62nd Annual Meeting of the Association for Computational Linguistics (Volume 1: Long Papers). 2024.

[2] Kwon, Yongchan, et al. "Datainf: Efficiently estimating data influence in lora-tuned llms and diffusion models." arXiv preprint arXiv:2310.00902 (2023).

[3] Pan, Yanzhou, et al. "Alinfik: Learning to approximate linearized future influence kernel for scalable third-parity LLM data valuation." Proceedings of the 2025 Conference of the Nations of the Americas Chapter of the Association for Computational Linguistics: Human Language Technologies (Volume 1: Long Papers). 2025.

---

> ### Author Rebuttal · Authors · 2026-03-31
>
> **Q1. It is unclear how the proposed approach differs from prior methods on influence estimation such as [1, 2].**\
> **A1.**  Thank you for the question. We agree that the distinction from prior influence-estimation methods should be clarified. Compared with [1], BLISS differs in both *goal* and *mechanism*. Lin et al. focus on *token-wise* influential training data retrieval, i.e., identifying training data that are influential to a given token prediction of an already trained/fixed LLM. Their method is essentially a *retrieval/attribution* framework at the token level. In contrast, BLISS is a sample-level *data selection* framework for pretraining.
>
> Compared with [2], BLISS also differs substantially. DataInf aims to efficiently approximate classical data influence, namely, the effect of upweighting or removing one training example on a model under a given fine-tuning state, using influence-function-style approximations. By contrast, BLISS does not approximate the marginal effect of a single example around a fixed model. Instead, it learns a *scoring model* through hypergradient-based bilevel optimization to capture the *dynamic* data preference of LLM.
>
> Therefore, unlike [1, 2], BLISS is not a classical influence estimator. It is a bilevel data selection method that targets the *dynamic* training influence of data for pretraining, rather than based on a fix model state.
>
> **Q2. Although the method is claimed to target LLM pre-training, the models used in the experiments appear relatively small (2.8B).**\
> **A2.** Thank you for the comment. We agree that our experiments are not at the frontier industrial scale. Our claim is not that BLISS has already been validated on the largest proprietary pre-training runs, but rather that it is designed for the *academic LLM pre-training* regime, where compute, infrastructure, and data access are substantially more limited.
>
> In fact, this setting is well aligned with recent discussions in the community. For example, the ICLR 2025 invited talk: Training Language Models in Academia: Challenge or Calling? (https://iclr.cc/virtual/2025/invited-talk/36784) explicitly emphasizes that language-model training is now largely industry-driven, while academia faces real constraints in compute, infrastructure, and proprietary data access; correspondingly, pretraining in academic budget is 1B model and 30B tokens (mentioned at 25:48 in the talk), which is consistent with our experiments.
>
> From this perspective, a 2.8B-scale model is not trivial in academia. Recent work on data selection [DsDm](https://arxiv.org/pdf/2401.12926) and [QuRating](https://arxiv.org/pdf/2402.09739) train the transformer with 1.3B parameters. Therefore, we believe our experimental scale is appropriate for evaluating methods intended for academic pre-training research.
>
> **Q3. The paper lacks comparisons with some recent baselines, such as Alinfik [3].**\
> **A3.** Thank you for pointing this out. We agree that ALinFiK is a relevant recent baseline and should be discussed more clearly. ALinFiK [3] computes (or approximates) *per-sample influence* directly on the *full target LLM* via training/test gradient inner products, followed by regression to fit an influence model. Even with gradient compression, these steps must be performed on the large LLM, making the approach *computationally intensive*.
>
> In contrast, BLISS never computes influence on the target LLM. Instead we learn sample importance via a *lightweight proxy model-based bilevel optimization*, and the lightweight proxy model makes hypergradient (need to compute hessian-inverse product) computation tractable. A token-level KL term with the frozen target LLM ensures the proxy reflects the target model’s data preference without requiring any gradients from the large model.
>
> In summary, ALinFiK computes influence on the target LLM itself, whereas BLISS performs influence learning through a lightweight proxy model, making bilevel *optimization efficient* while remaining *aligned* with the target model. We have cited these papers in related work and will provide more discussions in the revised version.

---

> > ### Author Rebuttal · Reviewer_BzUh · 2026-04-04
> >
> > Thank you for the rebuttal. Since my rating is positive, I would prefer to keep my current score.

---

> > > ### Author Response · Authors · 2026-04-05
> > >
> > > We sincerely appreciate your valuable feedback and thoughtful review. We would also be happy to address any further questions you may have.

---

### Official Review · Reviewer_2DtU · 2026-03-13

**Soundness:** 3
**Presentation:** 3
**Significance:** 3
**Originality:** 2
**Overall Recommendation:** 4
**Confidence:** 3

**Summary:**

This paper proposes BLISS, a bilevel data selection framework for LLM pretraining that learns a score model with a proxy model to estimate sample usefulness from scratch, without relying on external pretrained models. Experiments show that BLISS achieves comparable or slightly better downstream performance than state-of-the-art method while using slightly lower compute, and can reach the same performance faster in the 1B setting.

**Compliance With Llm Reviewing Policy:**

Affirmed.

**Final Justification:**

I appreciate the authors’ response. I consider this to be a borderline but overall positive work, and I will maintain my positive score of 4.

**Key Questions For Authors:**

See Weaknesses.

**Limitations:**

Yes

**Strengths And Weaknesses:**

**Strengths**
1. The motivation is clear and practically relevant.
2. The authors conducted extensive experiments to evaluate the proposed method.


**Weaknesses**
1. The methodological novelty is somewhat limited. The paper mainly combines existing ingredients such as bilevel optimization, proxy models, and influence-based data selection in a pretraining setting.
2. The empirical gains over MATES are positive but relatively modest. Average improvements are small, and several individual tasks do not improve, so the practical advantage is not yet fully compelling.
3. The claim that BLISS better captures long-term influence is plausible, but the evidence is still indirect. The experiments show effectiveness, but do not fully isolate whether the gains truly come from modeling longer-horizon effects.
4. Some key design choices are under-justified in the main paper, such as the validation set choice, top-20% selection ratio, and proxy reset strategy. It is therefore unclear how robust the method is to these choices.

---

> ### Author Rebuttal · Authors · 2026-03-31
>
> **Q1. The methodological novelty is somewhat limited.** \
> **A1.**  We want to highlight the main contributions of this paper. The first key contribution is the problem formulation: we formulate data selection as a bilevel optimization problem, where the upper level optimizes sample scoring for downstream validation set, and the lower level optimizes model parameter under weighted (scored) training data. The second key contribution is how we make this formulation practical for pretraining: instead of directly optimizing the target LLM in the lower level (computationally prohibitive), we use a lightweight proxy model, and explicitly constrain the output discrepancy of proxy model and the target LLM via KL alignment so that the proxy better tracks the target LLM's data preference and training behavior. This fidelity assumption of proxy model is empirically supported by our appendix domain-reweighting (Appendix J), where proxy-based and target-based trends are highly consistent. Third, unlike [MATES](https://arxiv.org/pdf/2406.06046) using one-step influence estimation, BLISS is designed to capture dynamic influence through training the lower level loss function for multiple steps rather than only one-step. Finally, we make the bilevel computation tractable with HVP-based hypergradient estimation; despite extra bilevel optimization, the overall cost remains practical and total FLOPs are lower than MATES in our main table (e.g., $8.08$ vs $8.11$ for 410M; $19.53$ vs $19.97$ for 1B, in $\times10^{19}$ FLOPs).
>
> **Q2.  The empirical gains over MATES are positive but relatively modest.**\
> **A2.**  We agree the average gain in 1B setting is modest (+$0.4\\%$ over MATES, $47.5\\%$ vs $47.9\\%$ in Table 4). However, the practical claim is about the quality-compute frontier, not winning every individual task: at nearly identical FLOPs, BLISS is better (410M: $8.08$ vs $8.11\times10^{19}$ FLOPs; 1B: $19.53$ vs $19.97\times10^{19}$ FLOPs) and achieves higher average accuracy at 1B ($47.9\\%$ vs $47.5\\%$). BLISS also wins most tasks against MATES (410M: 7/9; 1B: 6/9) and reaches the same performance $1.7\times$ faster in 1B (Figure 2 (b)). Moreover, our method improves over MATES by $0.6\\%$ when scaling up to 2.8B model pretraining.
>
> **Q3. The claim that BLISS better captures long-term influence is plausible, but the evidence is still indirect.**\
> **A3.** We agree that the phrase ``better captures long-term influence'' may be imprecise. What we intend to emphasize is that BLISS is **dynamic**, whereas methods such as [QuRating](https://arxiv.org/pdf/2402.09739) and [MATES](https://arxiv.org/pdf/2406.06046) are comparatively **static**. QuRating uses a scoring criterion derived from an external pretrained model before training, and MATES evaluates influence from the current model state via a one-step update. In contrast,
> BLISS dynamically updates the score model to produce sample weights, and proxy model is trained under the weight loss for multiple steps to an approximated optimal point, accordingly, while data preference signal evolving together with the target LLM during training.
>
> We will therefore revise the paper to replace the wording of ``long-term influence'' with a more precise statement that "BLISS captures the *dynamic data preference* of the target LLM".
>
> **Q4. Some key design choices are under-justified in the main paper.**\
> **A4.** For validation choice, We follow the prior work [DsDm](https://arxiv.org/pdf/2401.12926) and [MATES](https://arxiv.org/pdf/2406.06046)  to use LAMBADA  as a validation set. LAMBADA serves as an effective upper-level validation set because it provides a stable yet semantically demanding signal beyond local next-token prediction.  Empirically, LAMBADA gives the best average improvement (+$1.15\\%$ over random in the appendix C.4).
>
> We additionally experimented with a mixed validation set formed by combining multiple validation datasets (SQuAD, ARC-E, LAMBADAM, PIQA), and obtained an average downstream accuracy of $44.08\\%$ after 10B-token pretraining. When compared with the Average column in Figure 5 in appendix, this result is lower than the best single-validation-set choice, where LAMBADA achieves the highest average accuracy among all the other validation sets.
>
> For choice of the selection ratio, this hyperparameter is decided by the computational budget, since the selected data is used to pretrain the target LLM, which consumes most of the computational resources. This setting also follows the prior work MATES, as the computational budget limits the use of larger values for this hyperparameter.
>
> For proxy reset strategy, BLISS operates in rounds, where the training data $\xi$ distribution changes substantially from one round to another, which leads to a completely different lower-level problem. As a result, simply warm-starting from the proxy of the previous round can reduce the proxy's *plasticity*. See A1 in our response to Reviewer bj47 for more details.

---

> > ### Author Rebuttal · Reviewer_2DtU · 2026-04-04
> >
> > Thank you for the detailed response. My concerns have been adequately addressed, and I will maintain my original positive rating.

---

> > > ### Author Response · Authors · 2026-04-05
> > >
> > > We sincerely appreciate your valuable feedback and thoughtful review. We would also be happy to address any further questions you may have.

---

### Decision · Program_Chairs · 2026-04-30

**Decision:**

Accept (regular)

**Comment:**

This paper introduces BLISS, a novel bilevel optimization framework for data selection in Large Language Model (LLM) pretraining. Unlike existing methods that rely on expensive external pretrained oracle models, BLISS operates from scratch using a lightweight proxy model and a scoring model to estimate the dynamic influence of training samples.

Reviewers unanimously appreciated the clear motivation and practical relevance of removing the dependency on external oracles. The formulation of data selection as a bilevel optimization problem is technically sound and well-supported by thorough experiments. The empirical evaluation demonstrates that BLISS achieves competitive or slightly better downstream performance than state-of-the-art baselines (e.g., MATES) while providing notable computational speedups, particularly in the 1B parameter setting.

Moreover, The authors' rebuttal effectively addressed many concerns by providing finite-horizon convergence analysis and additional domain reweighting empirical clarifications. All reviewers reached a consensus, agreeing that the method is practically valuable and technically interesting. Therefore, the final recommendation is a Weak Accept.